# Scaling Linear Attention Capacity with Sparse State Expansion

**Yuqi Pan**[1,2†], **Yongqi An**[1,2†], **Zheng Li**[1†*], **Yuhong Chou**[3], **Ruijie Zhu**[4],
**Xiaohui Wang**[1], **Mingxuan Wang**[1], **Jinqiao Wang**[2], **Guoqi Li**[2*]
[1]ByteDance Seed  [2]Institute of Automation, Chinese Academy of Sciences
[3]The Hong Kong Polytechnic University  [4]UC Santa Cruz
lizheng.m@bytedance.com, guoqi.li@ia.ac.cn

## Abstract

The Transformer architecture, despite its widespread success, struggles with long-context scenarios due to quadratic computation and linear memory growth. While various linear attention variants mitigate these efficiency constraints by compressing context into fixed-size states, they often degrade performance in tasks such as in-context retrieval and reasoning. To address this limitation and achieve more effective context compression, we propose two key innovations. First, we introduce a **row-sparse update formulation** for linear attention by conceptualizing state updating as information categorization. This enables sparse state updates via softmax-based top-$k$ row selection, thereby extending receptive fields and reducing information interference. Second, we present **Sparse State Expansion (SSE)** within the sparse framework, which expands the contextual state into multiple partitions, effectively decoupling parameter size from state capacity while maintaining the sparse row-selection paradigm. Supported by efficient parallelized implementations, our design achieves highly discriminative state representations. We extensively validate SSE in both pure linear and hybrid (SSE-H) architectures across language modeling, in-context retrieval, and mathematical reasoning benchmarks. SSE demonstrates strong retrieval performance and scales favorably with state size. Moreover, after reinforcement learning (RL) training, our 2B SSE-H model achieves **state-of-the-art** mathematical reasoning performance among small reasoning models, scoring 64.5 on AIME24 and 50.2 on AIME25, significantly outperforming similarly sized open-source Transformers. These results highlight SSE as a promising and efficient architecture for long-context modeling.

## 1 Introduction

The Transformer architecture (Vaswani et al., 2017) has achieved remarkable success in various sequence modeling tasks, leveraging its attention mechanism for expressive modeling and high parallelism. However, it inherently struggles with processing long sequences due to quadratic computational complexity and linearly growing memory usage of key-value (KV) caches during inference.

To address these limitations, numerous linear attention variants have been proposed (Katharopoulos et al., 2020; Yang et al., 2024b; Dao & Gu, 2024; Yang et al., 2024a). Such approaches typically achieve sub-quadratic complexity and enable partial parallelism in training, along with RNN-like constant memory usage during decoding. However, most methods compress contextual information into fixed-size state matrices (e.g., $128 \times d$), which often compromises performance on tasks such as in-context retrieval and reasoning (Arora et al., 2024a;b; Jelassi et al., 2024; Akyürek et al., 2024). This raises a key question: how can context compression be designed to balance modeling fidelity and computational efficiency in long-context scenarios?

Motivated by the pursuit of more effective context compression, we propose two key innovations. First, we introduce a row-sparse update formulation for linear attention by conceptualizing state updating as an information categorization process. Specifically, we treat the rows of contextual states as distinct latent categories, interpret the key vector mapping $\mathbf{k}_t = f(\mathbf{x}_t, \mathbf{W}_k)$ as a category assignment function, and employ a softmax-based top-$k$ mechanism to select rows. At each step, this formulation selectively updates only state rows corresponding to predicted categories. Both theoretical analysis and synthetic experiments validate that our approach achieves larger receptive fields and reduces inter-category interference. Second, to mitigate the memory capacity constraints caused

by limited state sizes, we extend the contextual state within the row-sparse update framework and propose Sparse State Expansion (SSE). SSE expands the state into $N$ partitions with shared attention parameters and utilizes a write-read gate for partition selection, followed by a softmax-based selection of state rows within partitions. This design effectively decouples parameter size from state capacity while maintaining the sparse row-selection paradigm, and empirical results indicate that the states of SSE achieve lower inter-row similarity and higher singular value entropy, which enables more discriminative state representations. We also propose efficient parallelized implementations of SSE, employing masking and the varlen technique optimized for various training contexts.

We extensively validate SSE under both linear and hybrid architectures (denoted as SSE-H), across three core capabilities: language modeling, in-context retrieval, and mathematical reasoning. SSE demonstrates strong language modeling performance among advanced linear models (Yang et al., 2024b;a; Du et al., 2025), while SSE-H outperforms both hybrid and Transformer baselines. On long-context retrieval tasks, SSE consistently improves upon other linear attention models, and its hybrid variant significantly narrows the gap with softmax attention Transformers. For reasoning, 2B SSE-H model achieves scores of 64.5 on AIME24 and 50.2 on AIME25 with advanced RL training (Shao et al., 2024; Yu et al., 2025)—exceeding the best reported results from similarly sized open-source Transformers. Moreover, SSE exhibits strong scalability with respect to state capacity, and supports efficient conversion from pretrained Transformers at different scales, demonstrating its flexibility across model architectures and training regimes.

Our main contributions are as follows:

- We introduce a **row-sparse state update framework** that conceptualizes state updating as information categorization. This framework enables sparse state updates through softmax-based top-$k$ row selection, supported by both theoretical and empirical analysis.

- Within the proposed framework, we introduce **Sparse State Expansion (SSE)**, an efficient state expansion mechanism designed to effectively manage parameter count and preserve the sparse row-selection paradigm. Furthermore, we develop parallelized implementations of SSE, suitable for various training contexts.

- We extensively validate SSE and the hybrid SSE-H across diverse training stages and benchmarks. Our 2B hybrid model achieves **state-of-the-art reasoning performance** among small reasoning models, scoring 64.5 on AIME24 and 50.2 on AIME25—exceeding previous open-source softmax Transformers of comparable size.

## 2 PRELIMINARY

Autoregressive language models predict the next token by modeling the conditional probability $P_\theta(\mathbf{x}_t | \mathbf{x}_{<t})$. At inference, historical tokens $\mathbf{x}_{<t}$ are encoded into a compact contextual state $\mathbf{S}_t$ through different token-mixing mechanisms, facilitating sampling via $P_\theta(\mathbf{x}_t | \mathbf{S}_t) \triangleq P_\theta(\mathbf{x}_t | \mathbf{x}_{<t})$.

**Softmax Attention.** Softmax attention (Vaswani et al., 2017) stores the key-value vectors derived from historical tokens in a KV cache ($\mathbf{K}_t \in \mathcal{R}^{t \times d}, \mathbf{V}_t \in \mathcal{R}^{t \times d}$). As a new token $\mathbf{x}_t$ arrives, the corresponding vectors $\mathbf{k}_t, \mathbf{v}_t$ are appended to this cache. Attention is then computed by evaluating the interaction between the current token's query $\mathbf{q}_t$ and the KV cache:

$$\mathbf{S}_t = \{\mathbf{K}_t, \mathbf{V}_t\}, \tag{1}$$

$$\mathbf{K}_t = \begin{bmatrix} \mathbf{K}_{t-1} \\ \mathbf{k}_t \end{bmatrix}, \ \mathbf{V}_t = \begin{bmatrix} \mathbf{V}_{t-1} \\ \mathbf{v}_t \end{bmatrix}, \tag{2}$$

$$\mathbf{o}_t = \mathrm{softmax}(\mathbf{q}_t \mathbf{K}_t^\top) \mathbf{V}_t. \tag{3}$$

This append-based mechanism for state representation naturally enables high parallelism. However, as historical information is neither compressed nor pruned, it incurs efficiency bottlenecks for long sequences: memory and computation grow linearly with sequence length per step, and quadratically when all queries are processed in parallel during training.

**Vanilla Linear Attention.** Vanilla linear attention (Katharopoulos et al., 2020) compresses historical context into a fixed-size state matrix ($\mathbf{S}_t \in \mathcal{R}^{c \times d}$) using an outer-product update, where $c$ is a predefined constant (e.g., 128) independent of the sequence length:

$$\mathbf{S}_t = \mathbf{S}_{t-1} + \mathbf{k}_t^\top \mathbf{v}_t, \quad \mathbf{o}_t = \mathbf{q}_t \mathbf{S}_t. \tag{4}$$

For long contexts, this compression achieves significantly higher inference efficiency, requiring only constant memory and computational cost per time step. However, when $c \ll t$, the lossy compression causes performance gaps compared to softmax attention, particularly in tasks such as in-context retrieval and reasoning. By reformulating the linear recurrence into a chunk-wise parallel form, linear attention enables hardware-efficient and sub-quadratic training (Sun et al., 2023; Qin et al., 2023b; Yang et al., 2024b; Dao & Gu, 2024).

Thus, effective contextual compression has become essential for long-sequence processing, yet existing linear attention mechanisms have not fully exploited this potential. To address this, we introduce two improvements: a novel row-sparse update formulation for linear attention by conceptualizing state updating as information categorization, and Sparse State Expansion (SSE), which efficiently augments state capacity within the row-sparse update framework.

# 3 ROW-SPARSE STATE UPDATE: AN INFORMATION CATEGORIZATION PERSPECTIVE

Understanding the operational mechanism of contextual states is crucial for developing effective compression methods. To this end, we adopt a recurrent outer-product update formulation that provides a unified view of linear attention modeling (Chou et al., 2024):

$$\mathbf{S}_t = \mathbf{\Lambda}_t \mathbf{S}_{t-1} + \phi(\mathbf{k}_t)^\top \mathbf{v}_t. \tag{5}$$

Here, $\mathbf{\Lambda}_t$ is the state transition matrix, and $\phi$ is the feature map embedding $\mathbf{k}_t$ into a finite-dimensional space. This framework highlights two key design directions:

(1) **The role of $\mathbf{\Lambda}_t$ in historical information management.** Many modern linear attention variants (Sun et al., 2023; Qin et al., 2022; Peng et al., 2023) introduce recency bias via exponential decay or input-dependent gating (Yang et al., 2024b; Gu & Dao, 2023; Dao & Gu, 2024), while more recent works introduce structured updates such as diagonal-plus-low-rank (DPLR) matrices derived from the delta-rule (Yang et al., 2024c;a; Siems et al., 2025).

(2) **The role of $\phi(\mathbf{k}_t)$ in new information processing.** Early designs sought to approximate the exponential kernel of softmax attention (Katharopoulos et al., 2020; Choromanski et al., 2020; Peng et al., 2021; Zhang et al., 2024a), whereas recent variants often rely on simple mappings such as identity, ReLU, or SiLU (Sun et al., 2023; Yang et al., 2024b; Chou et al., 2024; Yang et al., 2024c;a).

While structured state transitions have seen steady progress, the role of the key feature map in effective state updating remains less understood. In this section, we focus on this second direction, proposing a row-sparse update formulation that conceptualizes state updating as information categorization. This is implemented via a top-$k$-then-softmax strategy for row-selection.

## 3.1 TREATING STATE ROWS AS LATENT CATEGORIES

In Equation 5, we treat state rows as distinct latent categories and conceptualize $\mathbf{k}_t = f(\mathbf{x}_t, \mathbf{W}_k)$ as a category assignment function[1], where larger values assign more information to the corresponding state rows. Consequently, each state row represents a distinct feature subspace, storing information that shares similar categorization outcomes. For a linear categorization function ($\mathbf{k}_t = \mathbf{x}_t \mathbf{W}_k$), we theoretically demonstrate that information assigned to the same state row exhibits similar features, as reflected by large inner products under this linear mapping:

**Proposition 1** *For inputs $\mathbf{x}_s$ satisfying $||\mathbf{x}_s||^2 = d$, and given the categorization rule $C^i = \{\mathbf{x}_s | \mathbf{x}_s \mathbf{W}_k^i > k_{th}^i\}$, where $k_{th}^i > 0$ is a category-specific threshold. For any inputs $\mathbf{x}_r, \mathbf{x}_s \in C^i$, the following inequality holds: $\mathbf{x}_r \mathbf{x}_s^\top > d \cos 2\theta^i$, where $\theta^i = \arccos(\frac{k_{th}^i}{\sqrt{d}||\mathbf{W}_k^i||})$, $s, r \in \{1, \dots, t\}$, and vice versa.*

See Proof 1 in Appendix D for the detailed proof. To validate this, we analyze the information assignments of linear attention using top-1 row-selection (Figure 9), and the information composition exhibits a clustering pattern, supporting our proposed categorization perspective. However, vanilla linear attention often suffers from inter-category mixing without this explicit top-$k$ hard assignment.

---

[1] For clarity, $\mathbf{k}_t$ denotes the vector after the $\phi$ mapping throughout the remainder of this paper.

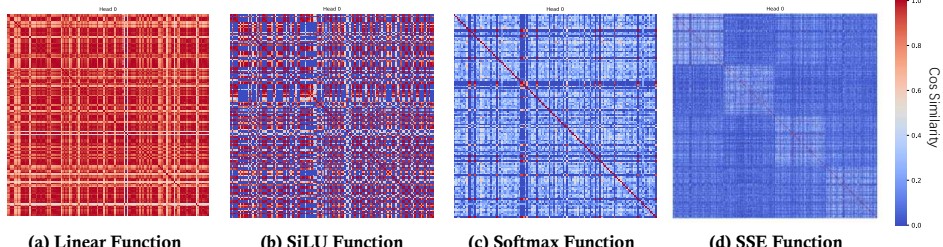

Figure 1: **Row-wise cosine similarity of contextual states in linear attention models with different categorization functions.** Cosine similarity is computed over the final states of multiple long-context inputs. Each plot shows a $128 \times 128$ similarity matrix across state rows; darker blue indicates lower similarity, reflecting more effective latent categorization.

This perspective allows us to reinterpret various designs of $f(\mathbf{x}_t, \mathbf{W}_k)$. Early models (Sun et al., 2023; Yang et al., 2024b) used a single linear transformation ($f(\mathbf{x}_t, \mathbf{W}_k) = \mathbf{x}_t \mathbf{W}_k$). Later works incorporated more expressive categorization functions, such as nonlinear activations (Yang et al., 2024c;a), gating mechanisms (Dao & Gu, 2024; Beck et al., 2024), and MLPs (Zhang et al., 2024a; Kasai et al., 2021). Beyond these examples, viewing state construction as information categorization also motivates novel architectural designs, including the softmax-based hard row-selection presented in the next section.

We evaluate different categorization strategies by analyzing row-wise cosine similarity of contextual states across models. As Figure 1(a)(b) shows, results highlight both the benefits of more expressive categorization functions and the limitations of existing linear attention—state rows exhibit substantial homogenization. We posit that this limitation arise from implicitly involving the notion of information categorization without fully exploiting category assignments for compressed storage.

### 3.2 ROW-SPARSE STATE UPDATE VIA TOP-$k$-THEN-SOFTMAX

In contrast to existing linear attention mechanisms, we explicitly leverage category assignments to derive a **row-sparse update formulation for linear attention**. From the perspective of latent categorization, we combine the top-$k$ hard assignment strategy with softmax categorization function:

$$\mathbf{k}_t = \text{softmax}(\text{top-}k(\mathbf{x}_t \mathbf{W}_k)), \quad \mathbf{S}_t = \tilde{\mathbf{\Lambda}}_t \mathbf{S}_{t-1} + \mathbf{k}_t^\top \mathbf{v}_t. \tag{6}$$

We apply the softmax only to the top-$k$ entries of $\mathbf{x}_t \mathbf{W}_k$, setting all remaining values to zero. Moreover, in gated linear attention variants such as GLA, the state transition matrix $\mathbf{\Lambda}_t$ incorporates a gating factor $\boldsymbol{\alpha}_t \in (0,1)$ (see Appendix C for details). In these cases, we eliminate the gates of non-selected rows using the same top-$k$ index set, yielding the modified transition matrix $\tilde{\mathbf{\Lambda}}_t$.

Using the linear categorization function as an illustration, we theoretically identify two fundamental limitations of existing linear attention designs together with the corresponding advantages of hard top-$k$ row selection:

(1) **Information interference (Propositions 2–3).** Updating all state rows introduces cross-category noise into memory, resulting in state row homogenization and reduced querying discrimination. In contrast, row-sparse updates encourage each state row to store similar information, yielding higher inter-row discriminability and more precise information organization.

**Proposition 2** *Under mild regularity conditions, the row-wise cosine similarity in vanilla linear attention, $\text{Cos}\left(\mathbf{S}_t^i, \mathbf{S}_t^j\right)$, is greater than that obtained under row-sparse updates, $\widetilde{\text{Cos}}\left(\mathbf{S}_t^i, \mathbf{S}_t^j\right)$.*

**Proposition 3** *For arbitrary queries $\mathbf{q}_t, \mathbf{p}_t \geq \mathbf{0}$, and states $\mathbf{S}_t$ with strictly bounded row norms, define the lower bound of the state-distinguishability measure as $\min_{\mathbf{p}_t, \mathbf{q}_t} \text{Cos}(\mathbf{q}_t \mathbf{S}_t, \mathbf{p}_t \mathbf{S}_t)$. Row-sparse updates reduce the lower bound compared to vanilla linear attention.*

(2) **Receptive fields (Proposition 4):** In gated variants of linear attention, uniform decay driven by strong recency bias restricts the effective receptive field. Row-sparse updates mitigate unnecessary decays, extending the receptive field over longer contexts and improving retrieval performance.

**Proposition 4** *Let $\mathbf{q}_t$ and $\mathbf{k}_s$ be all-ones vectors with $t \geq s > 0$. For gated linear attention with decay factors $\boldsymbol{\alpha}_t \in (0,1)$, the receptive field is upper-bounded: the attention score $p_{ts}$ falls below*

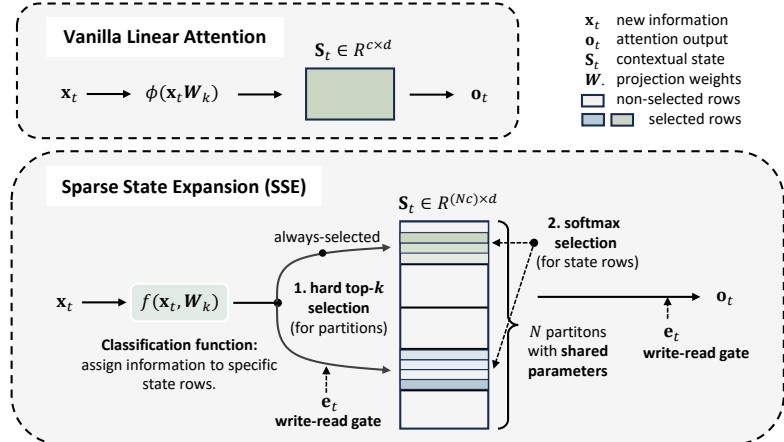

Figure 2: **Comparison between vanilla linear attention and SSE.** SSE expands the state into $N$ partitions within the row-sparse update framework, where a categorization function assigns information to specific state rows. All partitions share attention parameters. Sparse row selection follows two steps: (1) top-$k$ partition selection based on a write-read gate (blue indicates selected partitions; green marks an always-selected partition for training stability), and (2) row selection within the chosen partitions via softmax over key vectors.

*a given threshold $P_{th}$. In contrast, row-sparse update formulation can retain important information across arbitrary spans, ensuring the attention score remains above $P_{th}$.*

Detailed proofs are provided in Appendix D (Proofs 2–4). Synthetic MQAR (Arora et al., 2024a) experiments highlight the benefits of top-$k$ row-sparse updates for information recall (see Figure 7). Moreover, as Figure 1(c) illustrates, the low row-wise similarity exhibited by the states produced by the softmax-based row selector provides empirical support for the effectiveness of our framework.

To improve stability and mitigate category imbalance, we encourage a more uniform distribution of samples across state rows via an auxiliary loss (Fedus et al., 2022; Lepikhin et al., 2020): $\mathcal{L}_{balance} = \alpha \frac{c}{k} \sum_{i=1}^{c} f^i \cdot \mathbf{k}_t^i$, where $\alpha$ is a manually specified coefficient, $c$ the number of state rows, and $f^i$ the selection frequency of the $i$-th row. This loss complements the sparsity induced by top-$k$ row-selection and is optimized jointly with the main objective.

## 4 SSE: State Expansion under the Row-Sparse Framework

The most critical gap between linear and softmax attention lies in long-context retrieval and reasoning, a central focus of recent research. This primarily stems from the limited memory capacity caused by small state size. For instance, with $c = 128$, the memory budget of linear attention equals that of softmax attention with a context window of only 64 tokens. Therefore, increasing the contextual state size in linear models is key to addressing their current shortcomings. Recent works explore this issue from various perspectives (Arora et al., 2024b; Du et al., 2025; Guo et al., 2025b; Zhang et al., 2025). Building on this foundation, we pose the following question: *How can state expansion be designed from the perspectives of information categorization and sparse row-selection?*

### 4.1 Sparse State Expansion for Linear Attention

Within the row-sparse update framework, we expand the state rows by a factor of $N$, yielding $N \times c$ rows grouped into $N$ partitions. For each token $\mathbf{x}_t$, a top-$k$-then-softmax mechanism selects rows from the chosen partitions. Furthermore, to isolate the effect of state expansion, we fix the parameter size. This follows from the observation that softmax attention, with $4d^2$ parameters, can effectively manage unbounded states while maintaining strong in-context learning. Thus, in linear attention, the primary bottleneck lies in state size rather than parameter count.

Our design therefore aims to scale state capacity without increasing parameters while preserving sparse row selection. For efficiency, we avoid top-$k$ over all rows (Equation 6), instead applying hard top-$k$ selection over partitions followed by soft row selection within the chosen partitions using

softmax. With parameter sharing across partitions, this procedure is equivalent to applying softmax independently within each partition. Our core design consists of the following components:

**Shared attention parameters across partitions.** We use a single set of projection weights (QKV) across all partitions. This design choice, together with sparse partition selection, is equivalent to performing re-segmentation of input information along the sequence dimension, where each segment corresponds to a different partition, thereby enabling sparse token-level interactions. Additionally, parameter sharing mitigates training instability, such as overly sparse parameter updates.

**Write-read gate for hard partition selection, followed by soft row selection via softmax.** A gate vector $\mathbf{e}_t \in \mathcal{R}^N$ is first used to select the top-$k$ partitions, and within these partitions $\mathrm{softmax}(\mathbf{k}_t)$ further selects the state rows. Only the selected partitions are updated, and ablation results show that the softmax step yields over 40% row sparsity. Furthermore, the gate $\mathbf{e}_t$ is applied to both state input (KV) and output (Q), thereby governing both information writing and reading.

We refer to the linear attention variant that integrates the two core components described above as **Sparse State Expansion (SSE)**, illustrated in Figure 2. The complete procedure is as follows:

$$\mathbf{e}_t = \mathrm{softmax}(\mathbf{x}_t \mathbf{W}_e), \quad \mathcal{T} = \{i \mid \mathbf{e}_t^i \in \text{top-}k(\mathbf{e}_t)\}, \tag{7}$$

$$\mathbf{q}_t = \mathbf{x}_t \mathbf{W}_q, \ \mathbf{v}_t = \mathbf{x}_t \mathbf{W}_v, \ \mathbf{k}_t = \mathrm{softmax}(\mathbf{x}_t \mathbf{W}_k), \tag{8}$$

$$\mathbf{S}_t^i = \begin{cases} \boldsymbol{\Lambda}_t \mathbf{S}_{t-1}^i + \mathbf{e}_t^i \cdot \mathbf{k}_t^\top \mathbf{v}_t, & \text{for } i \in \mathcal{T} \\ \mathbf{S}_{t-1}^i, & \text{for } i \notin \mathcal{T} \end{cases} \tag{9}$$

$$\mathbf{o}_t = \sum_{i \in \mathcal{T}} \mathbf{e}_t^i \cdot \mathbf{q}_t \mathbf{S}_t^i. \tag{10}$$

(1) We compute a gate vector $\mathbf{e}_t \in \mathcal{R}^N$ via a linear projection and softmax, and use it for top-$k$ partition selection (Equation 7). (2) Next, we apply softmax across all rows of the selected partitions. Since partitions share the projection $\mathbf{W}_k$, this is equivalent to applying softmax within each partition, with the scaling factor absorbed into parameters (Equation 8). (3) To ensure the gate remains trainable, $\mathbf{e}_t$ is applied to both state input (KV) and output (Q) (Equations 9–10), thereby controlling both writing and reading and enabling more targeted optimization. (4) For partition $i$, the gated key is $\mathbf{k}_t^i = \mathbf{e}_t^i \cdot \mathrm{softmax}(\mathbf{x}_t \mathbf{W}_k)$ (Equation 9), which ensures normalization $\sum_i \sum_d (\mathbf{k}_t^i)_d = 1$. Only the selected partitions $\mathbf{S}_t^i$ are updated. (5) Within the write–read gate framework, we restrict state reading to the top-$k$ partitions using the same gate $\mathbf{e}_t$ (Equation 10).

Local interactions between consecutive tokens serve as an effective prior for language modeling (Fu et al., 2022; Peng et al., 2023; Arora et al., 2024b), which we leverage to stabilize the sparse component's training. To this end, we incorporate an always-selected partition and apply LoRA to QK projections, keeping the parameter count nearly constant. We further adopt a partition-based auxiliary loss in place of a row-based one. Leveraging parameter sharing and the sparsity induced by hard top-$k$ selection, SSE enables scaling up the state size of linear attention while keeping both computational and parameter overhead nearly constant.

Empirical analysis reveals that SSE's state partitions exhibit low inter-partition similarity; furthermore, state rows within each partition remain diverse, as illustrated in Figure 1(d). Overall, the state exhibits a diagonal pattern, indicating discriminative state representations. Furthermore, as Figure 3 shows, the singular value entropy of the SSE states (definition and discussion in Appendix F.1) is higher than that of GLA (Yang et al., 2024b). This indicates a more diverse and less compressible state composition, suggesting a more efficient utilization of state capacity in SSE. In addition, we analyze the receptive field of SSE and observe that it is significantly larger than that of GLA, indicating improved long-range information access (see Appendix F.1 for details).

## 4.2 Efficient Implementations of SSE

SSE expands the state into multiple partitions. During operator execution, we prioritize maintaining parallelism across partitions, avoiding sequential computation, which requires invoking the linear-attention kernel separately for each partition in a for-loop. Concurrently, we leverage sparsity to minimize unnecessary computational overhead. This section briefly outlines three practical implementation strategies, with further details and pseudocode provided in Appendix E.

**Naive implementation via masking.** For short and variable-length sequences, partitions are handled by replication and masking, enabling the use of larger chunk sizes. Specifically, activations are

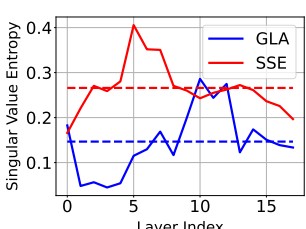 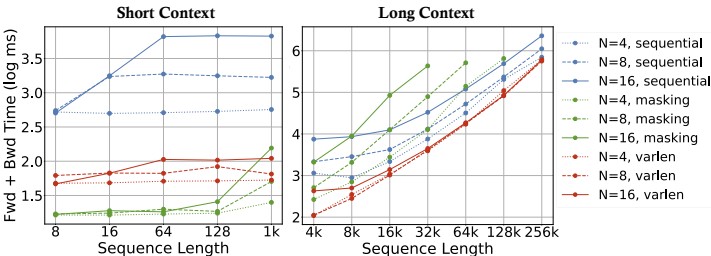

Figure 3: **Singular value entropy of states.** SSE with diagonal gating exhibits higher entropy than GLA.

Figure 4: **Speed comparison of different SSE implementations.** The number of selected partitions is fixed at $K = 1$, while the total number of partitions $N$ is varied. We set attention heads to 8, head dimension to 128, and cu_seqlens to $[0, L/2, L]$.

repeated $N$ times, and the top-$k$ indices are utilized to mask QKV vectors to zero. The additional dimension introduced by replication can be merged into the head dimension for execution.

**Efficient implementation via varlen technique.** For long contexts, redundant computation is avoided by first reordering QKV vectors with the top-$k$ indices, grouping tokens by partition. A new cu_seqlens parameter is then computed from the reordered sequences for operator execution. See Figure 8 for an illustration.

**Fusing always-selected partitions.** Always-selected partitions are fused through concatenation along appropriate dimensions, eliminating multiple sequential calls to the linear attention operator.

We evaluate the runtime performance of different SSE implementations, including sequential kernel invocation, masking, and varlen, across input lengths from 8 to 256k tokens. As Figure 4 illustrates, masking is competitive for short sequences ($\leq$1k), whereas varlen is more efficient for longer sequences. Furthermore, the varlen technique enables favorable scalability with respect to state size $N$, maintaining nearly constant runtime when $K$ is fixed.

## 5 EXPERIMENTS

We evaluate SSE across benchmarks spanning language modeling, in-context retrieval, and reasoning. All models are built on the MHA-SwiGLU backbone, with attention as the only varying component[2], and trained at two scales (600M and 2B parameters) under a unified pipeline covering pretraining, long-context extension, reasoning-focused distillation, reinforcement learning, and Transformer conversion. In Section 5.1, we show that SSE consistently outperforms existing linear attention models, with particularly strong gains in retrieval settings. When integrated into a hybrid architecture (-H), SSE achieves performance on par with Transformers. Notably, hybrid SSE-H demonstrates reasoning capabilities comparable to state-of-the-art Transformers under identical training regimes (Section 5.2). We primarily focus on SSE with GLA-style diagonal gating and additionally investigate integrating the delta-rule mechanism into SSE, denoted as SSE-GDN (Section 5.3). We further benchmark the efficiency of SSE in Section 5.4 and ablate several key design choices in Section 5.5. Additional details on architectures, baselines, and training setups are provided in Appendix F.

### 5.1 LANGUAGE MODELING AND RETRIEVAL

**Language Modeling.** We conduct small-scale pretraining to compare different attention mechanisms, using 600M and 2B models trained on 15B and 100B tokens, respectively. We then evaluate these models on zero-shot commonsense reasoning performance. As shown in Table 1, SSE-n4 achieves strong results among advanced linear attention models, outperforming Transformers and matching MoM despite using fewer parameters (600M vs. 800M). In the hybrid setting, SSE-H also surpasses both GLA-H and Transformers at the 2B scale, reaching state-of-the-art performance.

**In-context Retrieval.** SSE enhances retrieval through sparse state expansion and row-sparse updates. As Table 1 illustrates, SSE consistently outperforms other linear attention models on real-world retrieval-intensive tasks, narrowing the gap with Transformer. At both 600M and 2B scales,

---
[2]To ensure a fair comparison, all of our models vary only in the attention mixer and do not include convolutional layers—except for the Mamba family, which inherently uses 1D convolutions and unified blocks.

Table 1: **Zero-shot language modeling and recall performance.** For SSE and MoM, $n$ and $k$ (e.g., n4k1) denote expansion and sparsity ratios, respectively. Both adopt diagonal gating, as in GLA. MoM is implemented without token truncation for a stronger baseline. SSE-GDN uses DPLR gating, following GDN. The best and second-best results are bolded and underlined, respectively.

| Model | Wiki.↓ | CommonSense Reasoning | | | | | | | Real-world Recall | | | |
|---|---|---|---|---|---|---|---|---|---|---|---|---|
| | | PIQA | Hella. | Wino. | ARC-e | ARC-c | SIQA | Avg. | FDA | SWDE | SQuAD | Avg. |
| 600M params, 15B tokens | | | | | | | | | | | | |
| Transformer | 33.47 | 61.92 | 31.60 | 51.38 | 48.02 | 23.12 | 37.31 | 42.22 | 74.50 | 60.67 | 32.67 | 55.95 |
| GLA | 43.97 | 61.21 | 29.61 | 51.46 | 47.01 | 23.29 | 36.59 | 41.53 | 9.44 | 23.40 | 23.06 | 18.63 |
| Mamba | 36.62 | **63.82** | 32.20 | 50.43 | 49.62 | 23.38 | **38.59** | 43.01 | 6.35 | 21.60 | 24.93 | 17.63 |
| Mamba2 | 34.89 | 62.89 | **32.84** | 48.93 | 49.20 | 23.81 | 37.92 | 42.60 | 20.51 | 30.42 | 27.21 | 26.05 |
| GDN | 35.26 | 62.79 | 32.16 | **51.85** | 50.46 | 24.23 | 36.80 | **43.05** | 14.79 | 33.48 | 26.24 | 24.84 |
| KDA | 34.76 | 62.19 | 31.81 | 51.22 | 49.96 | 24.83 | 37.41 | 42.90 | 24.50 | 35.55 | 26.71 | 28.92 |
| **SSE-n4k1** | 35.52 | 61.59 | 31.71 | 51.62 | 49.45 | 24.74 | 38.33 | 42.91 | 33.67 | 33.30 | 26.51 | 31.16 |
| **SSE-n4k2** | 35.33 | 62.95 | 31.64 | 50.43 | 49.66 | **25.34** | 38.28 | **43.05** | 30.85 | 36.90 | 27.82 | 31.86 |
| **SSE-GDN-n4k1** | 34.15 | 63.11 | 32.22 | 50.36 | 50.59 | 23.98 | 37.46 | 42.95 | 41.56 | 41.13 | 30.83 | 37.84 |
| 800M (600M active) params, 15B tokens | | | | | | | | | | | | |
| MoM-n4k1 | 36.05 | 62.57 | 31.64 | 51.70 | 48.53 | 23.46 | 37.92 | 42.64 | 18.97 | 36.36 | 27.75 | 27.69 |
| MoM-n4k2 | 35.11 | 63.87 | 32.47 | 49.64 | 49.54 | 23.81 | 37.36 | 42.78 | 21.05 | 37.08 | 26.68 | 28.27 |
| 2B params, 100B tokens | | | | | | | | | | | | |
| Transformer | 16.46 | 71.82 | 52.70 | 58.80 | 63.76 | 31.31 | 42.89 | 53.55 | 86.48 | 82.99 | 49.53 | 73.00 |
| GLA | 21.30 | 68.44 | 42.82 | 54.38 | 59.43 | 28.24 | 41.45 | 49.13 | 51.27 | 59.86 | 36.73 | 49.29 |
| Mamba2 | 17.27 | 71.87 | 53.34 | 57.06 | 64.52 | 32.59 | 42.17 | 53.59 | 56.62 | 63.28 | 43.06 | 54.32 |
| GDN | 17.08 | **72.25** | 53.20 | 57.70 | 65.61 | 32.68 | **42.89** | 54.05 | 54.63 | 66.88 | 39.88 | 53.79 |
| **SSE-n4k1** | 16.92 | 71.98 | 53.62 | 60.14 | 66.67 | 32.94 | 42.07 | 54.57 | 71.23 | 69.94 | 43.20 | 61.46 |
| GLA-H | 17.27 | 70.95 | 51.58 | 57.30 | 65.03 | 31.31 | 42.73 | 53.15 | 78.22 | 78.49 | 45.84 | 67.52 |
| **SSE-H-n4k1** | 16.47 | 71.93 | 53.65 | 60.38 | 65.28 | 31.91 | 43.71 | 54.48 | 84.48 | 80.65 | 47.49 | 70.87 |

SSE-n4 significantly improves recall over GLA and GDN on all tasks, despite GDN's use of an explicit erasure mechanism. SSE-GDN further improves performance (see Section 5.3). While pure linear variants still lag behind Transformer, the hybrid SSE-H further boosts retrieval performance, surpassing GLA-H and approaching Transformer-level accuracy. We also benchmark our 2B models on synthetic S-NIAH tasks within the RULER (Hsieh et al., 2024) suite. As shown in Table 2, SSE demonstrates superior retrieval capabilities, further validating the benefits of its architectural design.

Table 2: **Performance comparison on Single-NIAH tasks in RULER.** All models have 2B parameters and are trained on 100B tokens using 8k variable-length sequences.

| Model | S-NIAH-1 | | | S-NIAH-2 | | | S-NIAH-3 | | |
|---|---|---|---|---|---|---|---|---|---|
| | 2K | 4K | 8K | 2K | 4K | 8K | 2K | 4K | 8K |
| Transformer | 100.0 | 100.0 | 100.0 | 100.0 | 100.0 | 100.0 | 100.0 | 100.0 | 100.0 |
| GLA | 100.0 | 100.0 | 87.6 | 100.0 | 81.8 | 23.2 | 88.0 | 62.2 | 16.2 |
| GDN | 100.0 | 100.0 | 100.0 | 98.8 | 62.4 | 8.2 | 96.2 | 63.8 | 9.0 |
| **SSE-n4k1** | 100.0 | 100.0 | 100.0 | 100.0 | 99.6 | 85.2 | 100.0 | 88.2 | 62.2 |
| GLA-H | 100.0 | 100.0 | 100.0 | 100.0 | 100.0 | 100.0 | 98.8 | 96.4 | 70.8 |
| **SSE-H-n4k1** | 100.0 | 100.0 | 100.0 | 100.0 | 100.0 | 100.0 | 100.0 | 99.4 | 97.4 |

**Data Scaling and Length Extension.** We scale both 2B SSE-H and Transformer baseline to 2T pre-training tokens, followed by a 32k context extension with 250B tokens. These models are then evaluated on benchmarks spanning knowledge, reasoning, and retrieval. As presented in Table 3, SSE-H remains competitive with the Transformer baseline across all task categories, while achieving higher average accuracy. Notably, it outperforms the Transformer on several challenging benchmarks, including MMLU, MMLU-Pro, and C-Eval, which are widely used to assess general knowledge and multilingual reasoning ability. We also report results on six single- and multi-needle tasks up to 32k from RULER, where SSE-H shows strong long-context retrieval performance (Appendix F.2, Table 10). These results highlight the robustness and scalability of our hybrid architecture under large-scale training and long-context settings.

**Converting Pre-trained Transformers.** Following prior work (Kasai et al., 2021), we convert a pretrained Transformer by replacing part of its softmax attention layers with linear attention, fol-

Table 3: **Benchmarking of 2B SSE-H and Transformer baseline after long-context extension.**

| Model | MMLU | MMLU-Pro | C-Eval | AGIEval | TrviaQA | BBH | SWDE | SQuAD | Drop | GSM8K | Avg. |
|---|---|---|---|---|---|---|---|---|---|---|---|
| Transformer | 52.6 | 24.2 | 55.9 | 38.9 | 21.8 | 38.8 | **85.5** | **52.0** | **30.9** | **50.6** | 45.1 |
| **SSE-H-n4k1** | **54.5** | **26.1** | **59.7** | **39.6** | **22.7** | **39.2** | 84.0 | 51.1 | 30.1 | 49.2 | **45.6** |

Table 4: **Reasoning ability of SSE-H.** Average accuracy (avg@32) for AIME24/25 is reported.

| Model | AIME24 | AIME25 | MATH500 | OlympiadBench | AMC23 |
|---|---|---|---|---|---|
| DeepScaleR-1.5B-Preview (Luo et al., 2025) | 43.1 | - | 87.8 | 50.0 | 73.6 |
| Qwen3-1.7B (Thinking) (Yang et al., 2025) | 48.3 | 36.8 | 93.4 | - | - |
| DeepSeek-R1-Distill-Qwen-1.5B (Guo et al., 2025a) | 28.9 | 23.5 | 83.9 | 43.3 | 62.9 |
| DeepSeek-R1-Distill-Qwen-7B (Guo et al., 2025a) | 55.5 | 39.2 | 92.8 | - | - |
| M1-3B (Wang et al., 2025b) | 23.0 | 22.0 | 81.7 | 43.6 | 56.0 |
| PromptCoT-Mamba-7B (Zhao et al., 2025) | 35.2 | 24.6 | 84.6 | 50.7 | - |
| Transformer-2B (Ours) | 64.1 | 52.8 | 93.0 | 83.3 | 92.0 |
| **SSE-H-n4k1-2B (Ours)** | 64.5 | 50.2 | 92.1 | 85.7 | 91.4 |

lowed by continued training. Applied to a 2B model during the 128k long-context training stage and evaluated on three 32k retrieval tasks, the converted SSE-H outperforms GLA-H and further narrows the gap with the original Transformer (Table 11, Appendix F.2).

## 5.2 REASONING ABILITY

Reasoning is particularly challenging for efficient architectures, as it demands multi-step computation, symbolic manipulation, and precise memory access. We evaluate whether hybrid linear models like SSE-H are effective for reasoning under standard pipelines with supervised distillation and reinforcement learning. We use math-focused benchmarks including MATH500, AIME24/25, AMC23, and OlympiadBench. For AIME24/25, we repeat the evaluation set 32 times and report avg@32 with temperature 1.0 and top-p 0.7, comparing against strong open-source small-scale reasoning models covering both softmax and hybrid designs.

As shown in Table 4, our 2B SSE-H-n4k1 achieves substantial gains over similarly sized open-source Transformers—improving AIME24 from 48.3 to 64.5 (+16.2) and AIME25 from 36.8 to 50.2 (+13.4)—and approaches much larger models such as DeepSeek-R1-Distill-Qwen-7B (Guo et al., 2025a). Trained under the same pipeline, SSE-H matches the Transformer baseline, and a 12B SSE-H variant converted from Transformer also maintains parity with its softmax counterpart (Appendix F.2, Table 9). These results show that SSE-H is competitive with state-of-the-art small reasoning models and scales effectively for reasoning tasks.

## 5.3 INTEGRATING SSE WITH GDN

Beyond GLA-style transitions, we additionally provide a comprehensive evaluation of integrating the delta-rule transition into SSE, denoted as SSE-GDN. Experimental details are provided in Appendix F.3. Overall, SSE-GDN achieves state-of-the-art recall performance (Tables 1 and 14), and its hybrid variant, SSE-GDN-H, consistently outperforms GDN-H in the 15B MoE setting (Table 15).

## 5.4 EFFICIENCY ANALYSIS

We benchmark the efficiency of SSE through attention runtime and end-to-end training throughput comparisons. Experimental details are provided in Appendix F.4. Overall, SSE surpasses Transformer at sequence lengths of 32k and beyond, while remaining slower than GLA and GDN due to the inherent overhead introduced by state expansion, reflecting a clear speed–performance tradeoff.

## 5.5 ABLATION STUDY

**Effectiveness of State Size Scaling in Recall.** We ablate 2B models (20B tokens) by varying the number of partitions ($n$) and the top-$k$ selection size ($k$). As shown in Figure 5, SSE scales recall effectively without increasing parameters: (1) with fixed sparsity ratio $k/n$, recall grows nearly linearly with $n$; (2) with fixed $n$, larger $k/n$ improves recall up to moderate levels (e.g., 50%), but saturates, reducing to vanilla linear attention when $k/n=1$. Since higher $n$ or $k/n$ increases computation and latency, we adopt the $n4k1$ setting for most experiments, leaving room for improvement.

**Effectiveness of Softmax-based Row Selection.** We ablate 600M models (15B tokens) to test softmax as the feature map for row-sparse updates. As shown in Table 5, softmax yields much higher

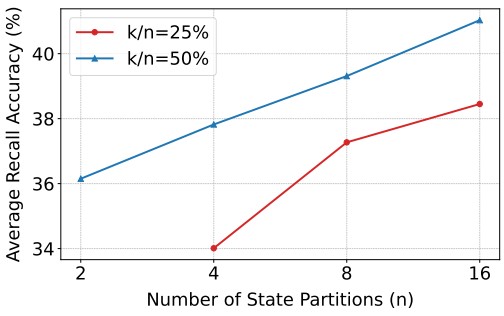
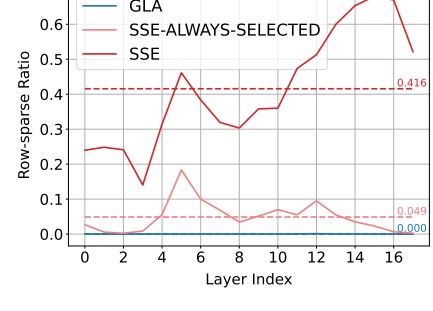

Figure 5: **Scalability of SSE with respect to state capacity.** $n$ denotes the number of partitions; $k$ is the top-$k$ hard selection size. Average recall accuracy is reported over FDA, SWDE, and SQuAD.

Figure 6: **Row sparsity ratio of state updating.** Measured as the fraction of $\mathbf{k}_t$ channels with values below 1e-5. For SSE, the sparsity is evaluated within the selected partitions.

Table 5: **Ablation on softmax row selection.**

| Models | Recall-Avg. |
|---|---|
| SSE-n4k1-k.silu | 24.23 |
| SSE-n4k1-k.softmax | 31.16 (+6.93) |

Table 6: **Ablation on shared parameters.**

| Models | Params. (non-embed) | Recall-Avg. |
|---|---|---|
| SSE-n4k1 | 600M (300M) | 31.16 |
| w/o shared-params | 890M (580M) | 25.95 (-5.21) |

recall than the common SiLU map, consistent with our information categorization perspective. We also analyze $\mathbf{k}_t$ sparsity in 2B GLA and SSE-n4k1 within the selected partitions (threshold 1e-5). Figure 6 shows that GLA with SiLU produces no sparsity, whereas SSE with softmax induces 5% and 42% sparsity, writing densely to the always-selected partition and sparsely to others.

**Effectiveness of Shared Attention Parameters.** SSE shares attention parameters (QKV and gate projection) across partitions, with LoRA applied only to QK of the shared one. We compare against a variant with separate parameters for all $N+1$ partitions ("w/o shared-params"). As shown in Table 6, despite nearly doubling non-embedding parameters, this variant yields worse recall and only minor CSR gains (see Table 13). This supports our core assumption: linear attention is limited by state capacity, not parameter count, and parameter sharing provides both efficiency and stability.

**Effectiveness of Write-Read Gating.** We ablate 2B models (100B tokens) to test four variants: no gate, write-only, read-only, and our proposed write–read gate (applied to both input KV and output Q). As shown in Table 7, the write–read gate consistently yields the best recall and commonsense reasoning (CSR) performance, validating its role in effective partition selection.

Table 7: **Ablation on write–read gating.** Comparison of four variants: no gate, write-only, read-only, and write–read (SSE). ✓indicates applied; ✗ indicates not applied.

| Models | Write | Read | Params. | Tokens | Wiki.↓ | FDA | SWDE | SQuAD | Recall-Avg. | CSR-Avg. |
|---|---|---|---|---|---|---|---|---|---|---|
| SSE-n4k1 | ✓ | ✓ | 2B | 100B | 18.44 | 64.07 | 65.17 | 40.65 | 56.63 | 53.63 |
| no gate | ✗ | ✗ | 2B | 100B | 18.63 | 51.91 | 63.91 | 39.81 | 51.87 (-4.76) | 53.01 (-0.62) |
| write gate | ✓ | ✗ | 2B | 100B | 18.65 | 54.54 | 61.12 | 37.10 | 50.92 (-5.71) | 52.82 (-0.81) |
| read gate | ✗ | ✓ | 2B | 100B | 18.62 | 57.26 | 64.45 | 40.18 | 53.96 (-2.67) | 52.91 (-0.72) |

## 6 CONCLUSION

We enhance the long-context modeling capacity of linear attention through two innovations: row-sparse state updates and Sparse State Expansion (SSE). Our framework conceptualizes state updates as information categorization and enables efficient expansion of contextual states into multiple partitions with controlled parameters, yielding scalable capacity and more discriminative representations. We demonstrate that SSE and its hybrid variant (SSE-H) achieve superior performance across language modeling, retrieval, and reasoning tasks. Notably, our 2B SSE-H achieves state-of-the-art mathematical reasoning among small-scale models, surpassing similarly sized Transformers. These results highlight SSE as a scalable and efficient architecture for high-fidelity long-context modeling.

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

## A    LIMITATIONS AND FUTURE WORK

While our experiments demonstrate the promising scalability of SSE with GLA-style transition mechanisms, we have not yet conducted comprehensive evaluations with delta-rule baselines (e.g., GDN). Moreover, more sophisticated categorization functions, such as state-aware mechanisms, could be effectively combined with SSE-GDN, offering a promising direction for future work.

Concurrently, we observe that non-hybrid SSE still exhibits a notable retrieval gap compared to well-trained Transformer models. Inspired by SSE's effective state size scalability, we consider further investigation into SSE's state scaling law a potential solution. Given the high search cost of partition count as a hyperparameter, we plan to utilize an up-training scheme to expand state size based on existing checkpoints. Thanks to our parameter sharing strategy, SSE can smoothly transition between different $N, K$ configurations, quickly recovering general ability without requiring extensive retraining or distillation. We consider SSE's state up-training a key direction for future work.

We acknowledge that SSE's current context-only hard-selection mechanism may not be optimal and warrants broader exploration. The computation of $\mathbf{e}_t$ provides a flexible interface for incorporating inductive biases into the model. A compelling future direction involves incorporating dependencies on positional and historical information, leading to a more general form such as $\mathbf{e}_t = g(t, \mathbf{x}_t, \mathbf{S}_{t-1})$.

## B    RELATED WORK

**Linear Attention.**    Linear attention (Katharopoulos et al., 2020) replaces the softmax kernel of Transformer attention with a dot-product of feature maps, thereby achieving linear complexity and more efficient long-context inference. Early research in this area (Choromanski et al., 2020; Peng et al., 2021; Choromanski et al., 2021; Zhang et al., 2024a) primarily focused on designing specialized non-negative feature maps to approximate the behavior of the softmax kernel. Subsequent works (Sun et al., 2023; Qin et al., 2023a) further simplified linear attention, removing unnecessary normalizer terms and other specialized feature maps. Concurrent with the evolution of linear attention, other generalized linear recurrent mechanisms emerged, such as State Space Models (SSMs) Gu et al. (2021; 2022); Fu et al. (2022); Gu & Dao (2023); Dao & Gu (2024) and linearized RNNs (Qin et al., 2023c; Peng et al., 2023; Orvieto et al., 2023; Beck et al., 2024). Through continuous development, these mechanisms have converged towards a unified decay and outer-product update formulation (Chou et al., 2024), similar to the unnormalized linear attention discussed in our work.

Recent findings (Yang et al., 2024b; Gu & Dao, 2023) highlight the benefit of introducing input-dependency into state decay, enabling more adaptive memory control through gating mechanisms. More recent trends involve densifying the state transition matrix, exemplified by delta-rule-based attentions (Yang et al., 2024c;a; Peng et al., 2025; Siems et al., 2025). While sophisticated structures, such as various TTT variants (Sun et al., 2024; Behrouz et al., 2024; Wang et al., 2025c; Behrouz et al., 2025) designed based on online learning, offer advanced state updates, their nonlinear dependencies often introduce trade-offs in implementation efficiency. In contrast to these works that primarily focus on the state transition matrix, our work re-examines how new information is more effectively incorporated into the state. We achieve this by innovatively connecting the state update mechanism to an information categorization process.

**State Expansion.** Recent literature (Arora et al., 2024a;b; Jelassi et al., 2024; Akyürek et al., 2024; Waleffe et al., 2024) consistently highlights the challenges posed by limited state size in linear attention models, particularly evident in retrieval tasks. Consequently, state expansion has emerged as a seemingly indispensable direction. Beyond directly expanding the state, Log-Linear Attention (Guo et al., 2025b) adopts a Fenwick tree structure to enable logarithmically growing state sizes. Our work bears the closest resemblance to Mixture-of-Memories (MoM) (Du et al., 2025), which models the state using a Mixture-of-Experts (MoE) approach and shares a similar sparse spirit with SSE. However, SSE designs a sparse state expansion scheme specifically from an information categorization perspective, explicitly decouples state capacity from parameter count, and introduces improved feature mapping and gating mechanisms. Furthermore, we provide a comprehensive theoretical framework demonstrating that SSE's state achieves more effective information storage. Our extensive experimental pipeline, spanning pretraining to RL, robustly confirms the proposed model's stability.

**Hybrid Models and Reasoning.** Beyond solely expanding the state of linear attention, a common compromise involves hybridizing some softmax attention layers. This approach sacrifices a degree of computational efficiency in exchange for significant improvements in retrieval performance. Recent works have explored both inter-layer (Lieber et al., 2024; Glorioso et al., 2024; Yang et al., 2024c; Li et al., 2025a; Wang et al., 2025a) and intra-layer (Dong et al., 2024; Li et al., 2025b; Behrouz et al., 2024) hybrid paradigms. Notably, such hybrid architectures have recently demonstrated promising overall performance, even at large scales (Li et al., 2025a; Chen et al., 2025; Liu et al., 2025). Concurrently, the reasoning capabilities of pure linear attention models remain relatively underexplored, largely relying on hybrid architectures (Wang et al., 2025b; Liu et al., 2025; Chen et al., 2025; Zhao et al., 2025). Our work is the first to demonstrate that, under identical pre-training paradigms, hybrid linear models can achieve reasoning performance on par with Transformer models, thereby enjoying the advantages of test-time scaling. Through fine-tuning and advanced RL training, we report that our 2B SSE-hybrid model exhibits exceptional mathematical reasoning ability, achieving state-of-the-art performance at its scale.

**Sparse MoE for Quadratic Attention.** In line with the Mixture-of-Experts (MoE) paradigm widely used in feed-forward networks (FFNs), dynamic computation can also be enabled in quadratic softmax attention through routing mechanisms. Existing approaches largely fall into two categories: (1) applying MoE to attention projection weights (e.g., $\mathbf{W}_q, \mathbf{W}_k, \mathbf{W}_v, \mathbf{W}_o$) (Fedus et al., 2022; Csordás et al., 2024), which closely parallels FFN-style MoE; and (2) performing MoE at the attention-head level by selecting a token-dependent subset of heads for computation (Jin et al., 2024; Piekos et al., 2025; Fu et al., 2024). For the latter, attention heads and their associated KVs can be viewed as a coarse form of state allocation. In contrast, the write–read gating used for top-$k$ partition selection in SSE extends this idea to the linear-attention setting, leveraging parameter sharing to achieve both computational and parameter efficiency. Moreover, softmax-based row selection can be interpreted as a finer-grained MoE mechanism operating at the level of individual state rows.

## C  FORMULATION OF LINEAR ATTENTION VARIANTS

We first present a recurrent outer-product update formulation that provides a unified view of various linear attention modeling (Chou et al., 2024):

$$\mathbf{S}_t = \mathbf{\Lambda}_t \mathbf{S}_{t-1} + \phi(\mathbf{k}_t)^\top \mathbf{v}_t. \tag{11}$$

Here, $\mathbf{\Lambda}_t$ denotes the state transition matrix, and $\phi$ is the feature map embedding $\mathbf{k}_t$ into a finite-dimensional space.

Gated linear attention variants (Sun et al., 2023; Qin et al., 2022; Peng et al., 2023) introduce recency bias via fixed exponential decay or input-dependent gating (Yang et al., 2024b; Gu & Dao, 2023; Dao & Gu, 2024). Specifically, gating mechanisms often employ diagonal (or scalar) matrices, $\mathbf{\Lambda}_t = \mathrm{diag}(\boldsymbol{\alpha}_t)$, where $\boldsymbol{\alpha}_t$ is input-dependent with values in $(0, 1)$:

$$\mathbf{S}_t = \mathrm{diag}(\boldsymbol{\alpha}_t)\mathbf{S}_{t-1} + \phi(\mathbf{k}_t)^\top \mathbf{v}_t. \tag{12}$$

Another line of work introduces structured updates such as diagonal-plus-low-rank (DPLR) matrices derived from the delta-rule (Schlag et al., 2021; Yang et al., 2024c;a; Siems et al., 2025). General DPLR matrices are typically structured as $\mathbf{\Lambda}_t = \mathrm{diag}(\boldsymbol{\alpha}_t) + \mathbf{a}_t^\top \mathbf{b}_t$:

$$\mathbf{S}_t = (\mathrm{diag}(\boldsymbol{\alpha}_t) + \mathbf{a}_t^\top \mathbf{b}_t)\mathbf{S}_{t-1} + \phi(\mathbf{k}_t)^\top \mathbf{v}_t. \tag{13}$$

A notable special case is DeltaNet, where $\mathbf{\Lambda}_t = \mathbf{I} - \beta_t \mathbf{k}_t^\top \mathbf{k}_t$. Its gated variant GDN adopts $\mathbf{\Lambda}_t = \alpha_t(\mathbf{I} - \beta_t \mathbf{k}_t^\top \mathbf{k}_t)$.

## D  THEORETICAL ANALYSIS OF ROW-SPARSE STATE UPDATES

In this section, we present theoretical analyses and key insights related to the information categorization perspective and the row-sparse update framework. Our discussion primarily focuses on the case of a linear categorization function, namely:

$$\mathbf{k}_t = f(\mathbf{x}_t, \mathbf{W}_k) = \mathbf{x}_t \mathbf{W}_k, \tag{14}$$

$$\mathbf{S}_t = \mathbf{\Lambda}_t \mathbf{S}_{t-1} + \mathbf{k}_t^\top \mathbf{v}_t. \tag{15}$$

**The contextual state updates in linear attention act as an information categorization process.**
By treating state rows as distinct latent categories and conceptualizing $\mathbf{k}_t = \mathbf{x}_t \mathbf{W}_k$ as a categorization function, we can demonstrate that information assigned to the same state row exhibits similar features. In the linear function setting, information similarity is measured by the inner product. The assignment result for category $i$ is defined as $C^i = \{\mathbf{x}_s | \mathbf{x}_s \mathbf{W}_k^i > k_{th}^i\}$, where $k_{th}^i > 0$ is a category-specific threshold. Proposition 1 shows that information assigned to the same state row exhibits large mutual inner products.

**Proposition 1** *For inputs $\mathbf{x}_s$ satisfying $||\mathbf{x}_s||^2 = d$, and given the categorization rule $C^i = \{\mathbf{x}_s | \mathbf{x}_s \mathbf{W}_k^i > k_{th}^i\}$, inputs belonging to the same category (row) $C^i$ satisfy $\mathbf{x}_r \mathbf{x}_s^\top > d \cos 2\theta^i$, where $\theta^i = \arccos(\frac{k_{th}^i}{\sqrt{d}||\mathbf{W}_k^i||})$, $s, r \in \{1, \ldots, t\}$, and vice versa.*

**Proof 1** *The categorization rule implies inputs with high acceptance intensity $\mathbf{k}_i = \mathbf{x}_s \mathbf{W}_k^i$ belong to the $i$-th category (row), and the category-specific threshold $k_{th}^i > 0$.*

*If $\mathbf{x}_s \in C^i$, then*

$$\mathbf{x}_s \mathbf{W}_k^i = \sqrt{d} ||\mathbf{W}_k^i|| \cos \theta > k_{th}^i, \tag{16}$$

$$\Rightarrow \cos \theta > \frac{k_{th}^i}{\sqrt{d} ||\mathbf{W}_k^i||} = \cos \theta^i, \tag{17}$$

*which means all inputs from category $i$ have an angle with $\mathbf{W}_k^i$ less than $\theta^i$, where $k_{th}^i \in (0, \sqrt{d} ||\mathbf{W}_k^i||)$ and $\theta^i \in (0, \frac{\pi}{2})$. This is equivalent to saying that the angle between inputs of the same category $i$ is less than $2\theta^i$. Thus we obtain:*

$$\mathbf{x}_r \mathbf{x}_s^\top > ||\mathbf{x}_r|| \, ||\mathbf{x}_s|| \cos 2\theta^i = d \cos 2\theta^i, \tag{18}$$

*where $\mathbf{x}_r, \mathbf{x}_s$ are arbitrary inputs from category $i$.*

*Furthermore, if we set the threshold large enough to ensure that $\theta^i < \frac{r}{4}$, where $r = \min_{i,j} \theta(\mathbf{W}_k^i, \mathbf{W}_k^j)$, we can ensure that for inputs from different categories, $\mathbf{x}_r \mathbf{x}_s^\top \leq d \cos 2\theta^i$.*

$\square$

**Current linear attention suffers from noise and forgetting due to incomplete use of categorization assignments.** As shown in Proposition 1, vanilla linear attention implicitly incorporating the notion of information categorization, but does not fully leverage the resulting category assignments for compressed storage:

$$\mathbf{S}_t^i = \sum_{j=1}^c \sum_{s_j \in C_j} w_{s_j}^i \mathbf{x}_{s_j} \mathbf{W}_v. \tag{19}$$

This mixing of information from different categories leads to two major issues: (1) **Noise**: Propositions 2 and 3 show that state rows in vanilla linear attention exhibit high similarity due to inter-category interference, resulting in state row homogenization and reduced querying effectiveness. (2) **Forgetting**: Proposition 4 demonstrates that applying uniform decay to all state rows adversely limits the receptive field.

In contrast, by explicitly leveraging category assignments, we introduce a **row-sparse update formulation of linear attention**:

$$\mathbf{S}_t^i = \sum_{s_i \in C_i} w_{s_i}^i \mathbf{x}_{s_i} \mathbf{W}_v. \tag{20}$$

Through this hard assignments, each row of the contextual state stores similar information belonging to the same category, resulting in higher inter-row discriminability and more precise information organization. Simultaneously, each element undergoes fewer decay operations, effectively extending the receptive field over longer contexts. The top-$k$ operation introduced in this work is one of the methods to achieve row-sparse updates.

**Definition 1** *The row-wise cosine similarity in vanilla linear attention is defined as* $\mathrm{Cos}\left(\mathbf{S}_t^i, \mathbf{S}_t^j\right) = \frac{\langle \mathbf{S}_t^i, \mathbf{S}_t^j \rangle}{||\mathbf{S}_t^i|| \, ||\mathbf{S}_t^j||}$. *Similarly, the row-wise cosine similarity under the row-sparse update formulation is denoted as* $\widetilde{\mathrm{Cos}}\left(\mathbf{S}_t^1, \mathbf{S}_t^2\right)$ *for differentiation.*

**Proposition 2** *Assume that* $\mathbf{W}_v$ *is orthogonal and* $w_{s_j}^i \geq 0 \, (i, j \in 1, \ldots, c)$ *in Equation 19, the row-wise cosine similarity in vanilla linear attention is greater than that of the corresponding row-sparse version, i.e.,* $\mathrm{Cos}\left(\mathbf{S}_t^i, \mathbf{S}_t^j\right) > \widetilde{\mathrm{Cos}}\left(\mathbf{S}_t^i, \mathbf{S}_t^j\right)$.

**Proof 2** *We use binary-categories as an illustration, which can be generalized to multi-categories scenario. Let* $i = 1, j = 2, \overline{\mathbf{S}_t^i} = \sum_{s_1 \in C_1} w_{s_1}^i \mathbf{x}_{s_1} + \sum_{s_2 \in C_2} w_{s_2}^i \mathbf{x}_{s_2}$. *Because* $\mathbf{W}_v$ *is an orthogonal matrix, we have:*

$$\langle \mathbf{S}_t^1, \mathbf{S}_t^2 \rangle = \overline{\mathbf{S}_t^1} \mathbf{W}_v \mathbf{W}_v^\top \overline{\mathbf{S}_t^2}^\top = \langle \overline{\mathbf{S}_t^1}, \overline{\mathbf{S}_t^2} \rangle. \tag{21}$$

*Since multiplying by an orthogonal matrix does not change the norm of a vector, therefore:*

$$\mathrm{Cos}\left(\mathbf{S}_t^i, \mathbf{S}_t^j\right) = \mathrm{Cos}\left(\overline{\mathbf{S}_t^i}, \overline{\mathbf{S}_t^j}\right) \tag{22}$$

*We have demonstrated that all inputs from category* $j$ *have an angle with* $\mathbf{W}_k^j$ *less than* $\theta^j$ *(Proposition 1). For* $\mathbf{x}_{s_j} \in C_j$ *and* $w_{s_j}^i \geq 0$, *the positive weighted sum* $\sum_{s_j \in C_j} w_{s_j}^i \mathbf{x}_{s_j}$ *still have an angle with* $\mathbf{W}_k^j$ *less than* $\theta^j$, *due to the parallelogram law. Thus we can let* $\sum_{s_j \in C_j} w_{s_j}^i \mathbf{x}_{s_j} = \alpha_j^i \mathbf{x}_j^i$, *where* $\mathbf{x}_j^i \in C_j$ *and* $\alpha_j^i > 0$ *implies module changed* $(||\mathbf{x}_j^i||^2 = d)$.

*Then, for vanilla linear attention, we have* $\overline{\mathbf{S}_t^1} = \alpha^1 \mathbf{x}_1^1 + \beta^1 \mathbf{x}_2^1$ *and* $\overline{\mathbf{S}_t^2} = \alpha^2 \mathbf{x}_1^2 + \beta^2 \mathbf{x}_2^2$. *For notational simplicity, we use* $\alpha^i, \beta^i$ *representing* $\alpha_1^i, \alpha_2^i$, *respectively. Let* $\cos\theta = \min\{\cos 2\theta^1, \cos 2\theta^2\}$, *we have:*

$$\langle \mathbf{x}_1^1, \mathbf{x}_1^2 \rangle > d\cos 2\theta^1 \geq d\cos\theta, \tag{23}$$

$$\langle \mathbf{x}_2^1, \mathbf{x}_2^2 \rangle > d\cos 2\theta^2 \geq d\cos\theta, \tag{24}$$

$$\langle \mathbf{x}_1^1, \mathbf{x}_2^2 \rangle \leq d\cos\theta, \tag{25}$$

$$\langle \mathbf{x}_2^1, \mathbf{x}_1^2 \rangle \leq d\cos\theta, \tag{26}$$

$$||\overline{\mathbf{S}_t^i}||^2 = (\alpha^i)^2 d + (\beta^i)^2 d + 2\alpha^i \beta^i \langle \mathbf{x}_1^i, \mathbf{x}_2^i \rangle \leq d(\alpha^i + \beta^i)^2. \tag{27}$$

*Because the row index is symmetric, we suppose* $\langle \mathbf{x}_2^1, \mathbf{x}_1^2 \rangle \geq \langle \mathbf{x}_1^1, \mathbf{x}_2^2 \rangle$ *without loss of generality. Then we can obtain:*

$$\mathrm{Cos}\left(\overline{\mathbf{S}_t^1}, \overline{\mathbf{S}_t^2}\right) > \frac{(\alpha^1 \alpha^2 + \beta^1 \beta^2)(\cos\theta + \epsilon)d + (\alpha^1 \beta^2 + \beta^1 \alpha^2)\langle \mathbf{x}_1^1, \mathbf{x}_2^2 \rangle}{d(\alpha^1 + \beta^1)(\alpha^2 + \beta^2)}, \tag{28}$$

*where* $\epsilon$ *refers to a sufficiently small constant. For the corresponding row-sparse linear attention, each state row corresponds to a latent category, so the contextual state stores information from different categories separately. That is,* $\overline{\mathbf{S}_t^1} = \alpha \mathbf{x}_1^1$ *and* $\overline{\mathbf{S}_t^1} = \beta \mathbf{x}_2^2$. *The similarity of the row-sparse attention is computed by:*

$$\widetilde{\mathrm{Cos}}\left(\overline{\mathbf{S}_t^1}, \overline{\mathbf{S}_t^2}\right) = \frac{\langle \mathbf{x}_1^1, \mathbf{x}_2^2 \rangle}{d}. \tag{29}$$

*The proof is given by:*

$$d\cos\theta + d\epsilon > \langle \mathbf{x}_1^1, \mathbf{x}_2^2 \rangle, \quad \textit{(holds according to Equation 25)} \tag{30}$$

$$\Rightarrow (\alpha^1 \alpha^2 + \beta^1 \beta^2)(\cos\theta + \epsilon)d > (\alpha^1 \alpha^2 + \beta^1 \beta^2)\langle \mathbf{x}_1^1, \mathbf{x}_2^2 \rangle, \quad (\alpha^i > 0, \beta^i > 0) \tag{31}$$

$$\Rightarrow \mathrm{Cos}\left(\overline{\mathbf{S}_t^1}, \overline{\mathbf{S}_t^2}\right) > \widetilde{\mathrm{Cos}}\left(\overline{\mathbf{S}_t^1}, \overline{\mathbf{S}_t^2}\right). \tag{32}$$

*(holds according to Equations 28, 29, and 31)*

*This means the state cosine similarity of vanilla linear attention is larger than the corresponding row-sparse version.*

*For the multi-categories scenario, we can still utilize triangle inequality of norm $||\overline{\mathbf{S}_t^i}||$ and Proposition 1 to reach the same conclusion. Note that in this case, we can solely focus on the two most different categories, namely $\langle \mathbf{x}_i^i, \mathbf{x}_j^j \rangle = \min_{k,l} \langle \mathbf{x}_k^i, \mathbf{x}_l^j \rangle$, to complete the derivation, which is enough to prove Proposition 3.*

*Combining Equations 22 and 32, we derive:*

$$\mathrm{Cos}\left(\mathbf{S}_t^i, \mathbf{S}_t^j\right) > \widetilde{\mathrm{Cos}}\left(\mathbf{S}_t^i, \mathbf{S}_t^j\right) \tag{33}$$

$\square$

Finally, in Proposition 3 we demonstrate that a more precise contextual state (established in Proposition 2) enables queries to extract diverse information more effectively. For a well-designed categorization function $f(\mathbf{x}_t, \mathbf{W}_k)$, we expect the reading operation $\mathbf{q}_t \mathbf{S}_t$ to allow different queries to extract distinct information, a property we refer to as state distinguishability. Higher distinguishability, achieved via row-sparse updates, reflects a more expressive and structured state. In contrast, lower distinguishability, typically observed in vanilla linear attention, leads to state homogenization and inter-category interference, which in turn hampers the model's ability to generate accurate output representations.

**Definition 2** *The measure $\min_{\mathbf{p}_t, \mathbf{q}_t} \mathrm{Cos}(\mathbf{q}_t \mathbf{S}_t, \mathbf{p}_t \mathbf{S}_t)$, referred to as state distinguishability, quantifies the minimum cosine similarity between states outputs, where $\mathbf{q}_t, \mathbf{p}_t$ are arbitrary queries.*

**Proposition 3** *For arbitrary queries $\mathbf{q}_t, \mathbf{p}_t \geq \mathbf{0}$, assume that the row norms of $\mathbf{S}_t$ are strictly bounded. The lower bound of the state distinguishability measure, $\min_{\mathbf{p}_t, \mathbf{q}_t} \mathrm{Cos}(\mathbf{q}_t \mathbf{S}_t, \mathbf{p}_t \mathbf{S}_t)$, is given by $\min_{i,j} \langle \mathbf{S}_t^i, \mathbf{S}_t^j \rangle$. Moreover, row-sparse updates reduces this lower bound compared to vanilla linear attention.*

**Proof 3** *Considering the cosine similarity, we have:*

$$\begin{aligned} \mathrm{Cos}(\mathbf{q}_t \mathbf{S}_t, \mathbf{p}_t \mathbf{S}_t) &= \frac{\langle \mathbf{q}_t \mathbf{S}_t, \mathbf{p}_t \mathbf{S}_t \rangle}{||\mathbf{q}_t \mathbf{S}_t|| \, ||\mathbf{p}_t \mathbf{S}_t||} \\ &\geq \frac{\langle \mathbf{q}_t \mathbf{S}_t, \mathbf{p}_t \mathbf{S}_t \rangle}{(||\mathbf{q}_t|| \, ||\mathbf{S}_t||)(||\mathbf{p}_t|| \, ||\mathbf{S}_t||)} \\ &= \frac{1}{||\mathbf{S}_t||^2} \langle \overline{\mathbf{q}}_t \mathbf{S}_t, \overline{\mathbf{p}}_t \mathbf{S}_t \rangle, \end{aligned} \tag{34}$$

*where $\overline{\mathbf{q}}_t = \frac{\mathbf{q}_t}{||\mathbf{q}_t||}, \overline{\mathbf{p}}_t = \frac{\mathbf{p}_t}{||\mathbf{p}_t||}$. Then the lower bound of the state distinguishability measure is given by:*

$$\min_{\mathbf{p}_t, \mathbf{q}_t} \mathrm{Cos}(\mathbf{q}_t \mathbf{S}_t, \mathbf{p}_t \mathbf{S}_t) \geq \frac{1}{||\mathbf{S}_t||^2} \min_{\mathbf{p}_t, \mathbf{q}_t} \langle \overline{\mathbf{q}}_t \mathbf{S}_t, \overline{\mathbf{p}}_t \mathbf{S}_t \rangle. \tag{35}$$

*The right side of Equation 35 is:*

$$\begin{aligned} \min_{\mathbf{p}_t, \mathbf{q}_t} \langle \overline{\mathbf{q}}_t \mathbf{S}_t, \overline{\mathbf{p}}_t \mathbf{S}_t \rangle &= \min_{||\overline{q}_t||=||\overline{p}_t||=1} \langle \sum_{i=1}^c \overline{q}_t^i \mathbf{S}_t^i, \sum_{j=1}^c \overline{p}_t^j \mathbf{S}_t^j \rangle \\ &= \min_{||\overline{q}_t||=||\overline{p}_t||=1} \sum_{i,j} \overline{q}_t^i \overline{p}_t^j \langle \mathbf{S}_t^i, \mathbf{S}_t^j \rangle \\ &= \min_{i,j} \langle \mathbf{S}_t^i, \mathbf{S}_t^j \rangle \end{aligned} \tag{36}$$

*We assume the row norm of $\mathbf{S}_t$ is bounded, which means:*

$$\exists \, \epsilon \quad \text{s.t.} \quad \min_i ||\mathbf{S}_t^i|| \geq \epsilon. \tag{37}$$

*The assumption implies that the contextual state is numerically bounded, thus the inner-product between state rows is governed by the cosine similarity:*

$$
\begin{aligned}
\min_{\mathbf{p}_t, \mathbf{q}_t} \mathrm{Cos}(\mathbf{q}_t \mathbf{S}_t, \mathbf{p}_t \mathbf{S}_t) &\geq \frac{1}{||\mathbf{S}_t||^2} \min_{i,j} \langle \mathbf{S}_t^i, \mathbf{S}_t^j \rangle \\
&= \frac{1}{||\mathbf{S}_t||^2} \min_{i,j} \mathrm{Cos}\left( \overline{\mathbf{S}_t^i}, \overline{\mathbf{S}_t^j} \right) \cdot ||\mathbf{S}_t^i|| \cdot ||\mathbf{S}_t^j||, \quad \textit{(holds according to Defination 1)} \\
&\geq \frac{\epsilon^2}{||\mathbf{S}_t||^2} \min_{i,j} \mathrm{Cos}\left( \overline{\mathbf{S}_t^i}, \overline{\mathbf{S}_t^j} \right), \textit{(holds according to Equation 37)}
\end{aligned}
\tag{38}
$$

*which shows that the lower bound of the state distinguishability measure is related to the cosine similarity between the two most different categories. Because row-sparse update formulation has reduced the right-side similarity to $\widetilde{\mathrm{Cos}}\left( \overline{\mathbf{S}_t^i}, \overline{\mathbf{S}_t^j} \right)$, it can be seen as a way to reduce the lower bound of the measure.*

$\square$

In Proposition 4, we demonstrate that the exponential cumulative decay inherent to vanilla gated linear attention adversely affects receptive fields. In contrast, row-sparse update formulation mitigates this issue and can theoretically achieve infinitely long receptive fields, as sparse decay ensures the retention of important information.

**Definition 3** *The attention score of $\mathbf{x}_t$ with respect to $\mathbf{x}_s$ is defined as $p_{ts} = \mathbf{q}_t((\prod_{j=s+1}^t \boldsymbol{\alpha}_j) \odot \mathbf{k}_s)^\top$.*

**Definition 4** *Let $p_{ts}$ denote the attention scores and $P_{th}$ be a given threshold, where $t \geq s > 0$. The receptive field at time $t$ is defined as: $M_t = \max\{t - s \mid p_{ts} \geq P_{th}\}$.*

**Proposition 4** *Let $\mathbf{q}_t$ and $\mathbf{k}_s$ be all-ones vectors, and let $t \geq s > 0$. For vanilla gated linear attention with decay factors $\boldsymbol{\alpha}_t \in (0, 1)$, the receptive field is upper-bounded by $M_t = \max_{k=1,\dots,d} \log(\frac{P_{th}}{d}) / \log(\max_{s+1 \leq j \leq t} \boldsymbol{\alpha}_j^k)$, such that the attention score $p_{ts}$ is less than a given threshold $P_{th}$. Conversely, row-sparse update formulation can retain important information across arbitrary spans, ensuring the attention score remains above the threshold $P_{th}$, i.e., $M_t = t$.*

**Proof 4** *Assume that $\mathbf{q}_t = \mathbf{k}_s = [1, \dots, 1] \in \mathcal{R}^d$, then the attention score:*

$$
p_{ts} = \mathbf{q}_t(( \prod_{j=s+1}^t \boldsymbol{\alpha}_j) \odot \mathbf{k}_s)^\top = \sum_{k=1}^d \prod_{j=s+1}^t \boldsymbol{\alpha}_j^k,
\tag{39}
$$

*which can characterize the utility of only decay.*

*Effective receptive field requires the attention score $p_{ts} \geq P_{th}$. The corresponding necessary condition is:*

$$
\exists \, k, \quad s.t. \quad (\max_{s+1 \leq j \leq t} \boldsymbol{\alpha}_j^k)^{t-s} \geq \frac{P_{th}}{d}.
\tag{40}
$$

*That is,*

$$
\exists \, k, \quad s.t. \quad t - s \leq \log(\frac{P_{th}}{d}) / \log(\max_{s+1 \leq j \leq t} \boldsymbol{\alpha}_j^k)
\tag{41}
$$

*Therefore, $M_t = \max_{k=1,\dots,d} \log(\frac{P_{th}}{d}) / \log(\max_{s+1 \leq j \leq t} \boldsymbol{\alpha}_j^k)$ represents the upper bound of receptive fields. For earlier tokens at time $s'$, if the receptive field exceeds this upper bound ($t - s' > M_t$), the information is forgotten as $p_{ts} < P_{th}$.*

*In contrast, for row-sparse update variant of linear attention whose decay values can be scattered to actually equal 1, we can ensure the retention of important information across arbitrary spans by simply considering:*

$$
\exists \, k, \quad s.t. \quad \boldsymbol{\alpha}_j^k = 1 \text{ for } \forall s + 1 \leq j \leq t.
\tag{42}
$$

*This implies that row-sparse update linear attention can ensure $M_t = t$.*

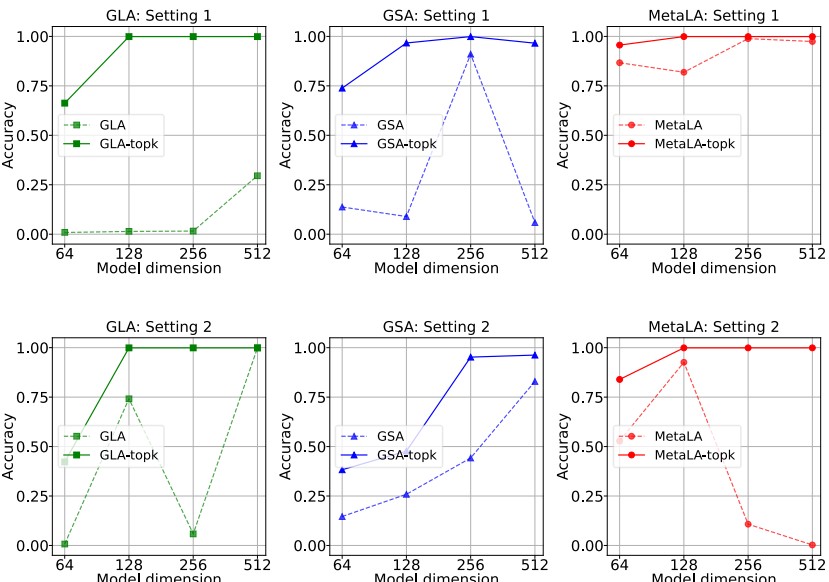

Figure 7: **MQAR results of linear attention models and their top-$k$ row-sparse variants.** Settings 1 and 2 introduce zero tokens and random noise tokens, respectively. Top-$k$ row-sparse updates lead to longer effective context and more accurate information storage within the state. Moreover, with row-sparse updates, recall performance increases monotonically with respect to state size (i.e., model dimension).

$\square$

**Synthetic experiment validation.** We validate our theoretical analysis by applying top-$k$ sparsity during the state update process of three linear attention models, and evaluate recall performance on the synthetic MQAR task (Arora et al., 2024a). Specifically, only the state rows corresponding to the $k$ largest values in $\mathbf{k}_t$ are updated at each step. In MQAR (Multi-Query Associative Recall), the model must retrieve previously stored key-value pairs in response to multiple queries. We evaluate recall under two settings: (1) In the first setting, zero tokens are inserted between key-value pairs and queries to explicitly evaluate the model's effective receptive field. (2) In the second setting, random noise tokens are added, testing the model's robustness to interference during memory retention.

We conduct experiments on GLA (Yang et al., 2024b), MetaLA (Chou et al., 2024), and GSA (Zhang et al., 2024b)—three representative linear attention variants. These models respectively incorporate gated updates, decay-key coupling, and two-pass recurrence. We set $k = d/4$, and train 2-layer models with alternating token and channel mixers, using 2 attention heads. As shown in Figure 7, the results demonstrate that row-sparse updates offer a simple and general approach for significantly enhancing the recall capabilities of vanilla linear attention, leading to longer effective context and more accurate information storage within the state. Moreover, with row-sparse updates, recall performance increases monotonically with respect to state size (i.e., model dimension).

# E    DETAILS OF EFFICIENT SSE IMPLEMENTATIONS

SSE expands the contextual state into multiple partitions, each consisting of different subsets of tokens. During operator execution, our goal is to preserve parallelism across partitions, rather than relying on sequential invocation. To minimize unnecessary computational overhead, we introduce two implementations tailored to different sequence length regimes: a masking-based version for short sequences, and a varlen-based version for long sequences. In addition, we describe a simple fusion strategy for managing always-selected partitions.

**Naive implementation via masking for short contexts.** In the chunk-wise operator of linear attention, each kernel instance is responsible for processing a single data chunk. To improve execution

efficiency, the chunk size is typically set to a power of two (e.g., 32, 64, or 128), which facilitates memory alignment and GPU-level parallelism. The optimal chunk size depends on both the target GPU architecture and the specific characteristics of the operator, and is typically determined through empirical tuning. However, variable-length training often includes a large proportion of short sequences, which can lead to per-partition lengths falling below the optimal chunk size, or even below the minimum threshold (e.g., 16). In such cases, the varlen implementation incurs additional operational overhead, and grouping tokens by their assigned partitions and processing them with cu_seqlens can become inefficient. To address this, we adopt a masking-based strategy that increases parallelism through replication while enabling the use of larger chunk sizes. Specifically, each activation is replicated across partitions, and masking is applied based on the top-$k$ selection to ensure that each partition only attends to its assigned tokens. This approach improves GPU utilization without requiring token reordering. The corresponding pseudocode is presented in Algorithm 1.

---

**Algorithm 1:** SSE Implementation via Top-$k$ Masking

**Input:** $\mathbf{Q}, \mathbf{K}, \mathbf{V} \in \mathbb{R}^{L \times H \times D}$, $\mathbf{E} \in \mathbb{R}^{L \times N}$, cu_seqlens $\in \mathbb{R}^{S}$, Number of partitions $N$, Top-k partitions $K$

**Output:** $\mathbf{O} \in \mathbb{R}^{L \times H \times D}$

    // Compute top-$k$ selection mask

1  $\mathbf{M}_{\text{topk}} = \text{TopkAndMask}(\mathbf{E}, K, \dim = 1) \in \{0, 1\}^{L \times N}$

    // Repeat input across partitions and apply masking

2  $\mathbf{Q}, \mathbf{K}, \mathbf{V} = \text{Repeat}(\mathbf{Q}, \mathbf{K}, \mathbf{V}, N, \dim = 1) \in \mathbb{R}^{L \times N \times H \times D}$;

3  $\mathbf{Q} = \mathbf{Q} \odot \mathbf{M}_{\text{topk}}$ ;

4  $\mathbf{K} = \mathbf{K} \odot \mathbf{M}_{\text{topk}}$ ;

5  $\mathbf{V} = \mathbf{V} \odot \mathbf{M}_{\text{topk}}$ ;

    // Rearrange for linear attention

6  $\mathbf{Q}, \mathbf{K}, \mathbf{V} = \text{Rearrange}(\mathbf{Q}, \mathbf{K}, \mathbf{V}, \dim = (1, 2)) \in \mathbb{R}^{L \times (NH) \times D}$;

    // Linear attention computation

7  $\mathbf{O} = \text{LinearAttention}(\mathbf{Q}, \mathbf{K}, \mathbf{V}, \text{cu\_seqlens})$;

8  **return** $\mathbf{O}$;

---

**Efficient implementation via varlen technique for long contexts.** During the long-context continual training phase, the sequence lengths are generally longer and more evenly distributed, allowing chunk-wise computation to operate only on the relevant tokens within each partition. This avoids the redundant computations over masked tokens inherent in the naive masking-based implementation. Specifically, we first derive the top-$k$ partition indices and use them to reorder the QKV vectors, grouping tokens sequentially by partition (from 1 to $N$) within each sample. Next, a new cu_seqlens is computed based on the reordered sequences and their corresponding partition assignments. At this stage, each resulting subsequence corresponds to a specific partition within a specific sample. Given this reordering and the updated cu_seqlens, all partitions can be processed in parallel using chunk-wise linear attention, without introducing additional computational overhead. This implementation exhibits favorable scalability with respect to state size $N$, maintaining nearly constant overhead as long as $K$ (the number of selected partitions) remains fixed. The corresponding pseudocode is presented in Algorithm 2.

---

**Algorithm 2:** SSE Implementation via Varlen Technique

**Input:** $\mathbf{Q}, \mathbf{K}, \mathbf{V} \in \mathbb{R}^{L \times H \times D}$, $\mathbf{E} \in \mathbb{R}^{L \times N}$, cu_seqlens $\in \mathbb{R}^{S}$, Number of partitions $N$, Top-k partitions $K$

**Output:** $\mathbf{O} \in \mathbb{R}^{L \times H \times D}$

    // Obtain reorder index $\mathbf{I} \in \mathbb{R}^{KL}$ and updated sequence offsets

1  $\mathbf{I}, \text{new\_cu\_seqlens} = \text{GetIndexAndOffsets}(\mathbf{E}, K, \text{cu\_seqlens})$

    // Group tokens by partition order

2  $\mathbf{Q}, \mathbf{K}, \mathbf{V} = \text{Reorder}(\mathbf{Q}, \mathbf{K}, \mathbf{V}, \mathbf{I}, \dim = 0) \in \mathbb{R}^{(KL) \times H \times D}$;

    // Linear attention computation with varlen partitioning

3  $\mathbf{O} = \text{LinearAttention}(\mathbf{Q}, \mathbf{K}, \mathbf{V}, \text{new\_cu\_seqlens})$;

4  **return** $\mathbf{O}$;

---

**1. Reorder inputs & compute new cu_seqlens**  **2. Linear attention computation**  **3. Reorder outputs**

Figure 8: **Varlen implementation of SSE (n4k1 as an example).** We first use top-$k$ partition indices to reorder the QKV vectors, grouping tokens by partition (from 1 to $N$) within each sample. A new `cu_seqlens` is then constructed from the reordered sequences, where each subsequence corresponds to a specific partition. All partitions are subsequently processed in parallel using linear-attention kernels. Finally, the outputs are reordered back to the original token order.

**Sequential kernel invocation via for-loop.** For comparative analysis, we present the vanilla sequential implementation. Initially, an additional dimension representing partitions is introduced, and tokens are gathered according to their assigned partitions. Subsequently, the chunk linear attention kernel is sequentially invoked over all partitions within a for-loop, bypassing varlen control. While this approach circumvents the overhead potentially introduced by varlen, it sacrifices inherent parallelism across partitions. Consequently, this method exhibits poor scalability: runtime increases significantly with larger values of $N$ due to its sequential computation.

**Fusing always-selected partitions.** Always-selected partitions can be fused through concatenation to avoid multiple sequential calls to the linear attention operator. In the naive implementation, always-selected partitions are concatenated along the head dimension, and no masking is applied to this portion. In the varlen implementation, always-selected partitions are concatenated with the reordered sequence along the sequence dimension, while segment-wise computation remains controlled via the cu_seqlens parameter. These modifications allow the linear attention operator to be invoked only once.

In the inference setting, the chunk-wise implementation described above applies directly to the pre-filling stage. During decoding, we leverage sparse indices to perform recurrent computation solely on the selected partitions, significantly reducing computational overhead.

## F EXPERIMENT DETAILS

**Model Configurations.** Model configurations for our experiments are summarized in Table 8. We build all models on the MHA-SwiGLU architecture (Touvron et al., 2023; Shazeer, 2020), varying only the attention mechanism to ensure fair comparisons. Accordingly, all models exclude convolutional layers, except for the Mamba family, which inherently employs 1D convolutions and unified blocks. We compare SSE against standard Transformer (using softmax attention) and several representative linear attention baselines, such as GLA (Yang et al., 2024b), GDN (Yang et al., 2024a), Mamba (Gu & Dao, 2023), Mamba2 (Dao & Gu, 2024), KDA (Team et al., 2025) and MoM (Du et al., 2025). Both SSE and MoM adopt GLA-style state transition mechanisms, with $n$ denoting the state expansion factor (i.e., the number of state partitions) and $k$ representing the top-$k$ hard selection size per token. The sparsity ratio is defined as $k/n$. We further investigate integrating the delta-rule mechanism into SSE, denoted as SSE-GDN. For the SSE family, we utilize a single always-selected partition and set the low-rank dimension to 64 for QK projections. The coefficient for the auxiliary loss is 0.01 across both SSE(-GDN) and MoM. Additionally, we also evaluate layer-wise hybrid architectures, constructed by adding one softmax attention layer after every five linear attention layers. In our 2B setting, this results in 3 softmax attention layers out of 18 total layers. These models are denoted by the "-H" suffix (e.g., SSE-H).

**Training Setup.** We conducted our primary experiments with two model scales: 600M and 2B parameters. The non-embedding parameters are 300M and 1.3B, respectively. During pretraining, all models are trained with AdamW, using a maximum sequence length of 8192 and a total batch size of 4M tokens. The learning rate schedule includes a linear warm-up phase followed by a constant

Table 8: **Model Architectures.**

| Model Size | Non-Embedding Params | Layers | Hidden Dimension | Heads |
|------------|---------------------|--------|------------------|-------|
| 600M | 300M | 24 | 1024 | 8 |
| 2B | 1.3B | 18 | 2304 | 18 |

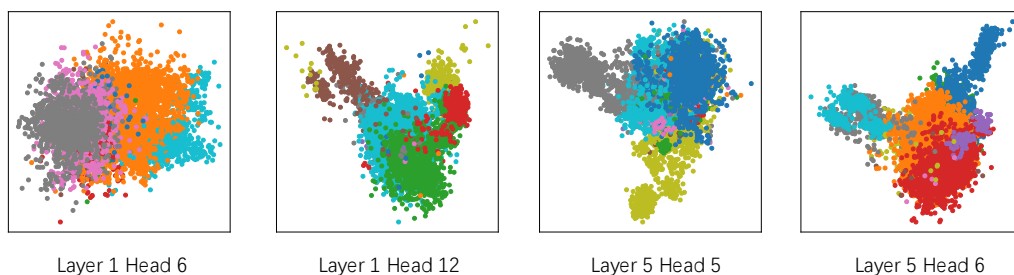

| Layer 1 Head 6 | Layer 1 Head 12 | Layer 5 Head 5 | Layer 5 Head 6 |

Figure 9: **Clustering of information within linear attention state rows.** We observe that learned state representations reveal clear clustering patterns. Specifically, we assign each token's value vector (represented as a point) to a specific state row (indicated by color) by taking the maximum activation over its corresponding key vector. This assignment demonstrates that information within the same row tends to share similar feature representations.

learning rate of 6e-4. We apply a weight decay of 0.1 and a gradient clipping of 1.0. Following pre-training, we perform context length extension up to 32k tokens, using cosine learning rate decay. Subsequently, we perform a reasoning-oriented data distillation stage using a fine-tuning set of approximately 80k examples, trained for 5 epochs. Finally, we apply reinforcement learning with GRPO (Shao et al., 2024) for 230 steps. During RL training, we sample 8 responses per prompt with a generation limit of 32k tokens.

## F.1 ANALYSIS OF CONTEXTUAL STATES

**Singular Value Entropy.** The singular value entropy (Roy & Vetterli, 2007) is computed by:

$$H = -\frac{1}{\log n} \sum_{i=1}^{n} \frac{\sigma_i^2}{\sum_{j=1}^{n} \sigma_j^2} \log \frac{\sigma_i^2}{\sum_{j=1}^{n} \sigma_j^2}, \tag{43}$$

where $\sigma_i$ (for $i \in \{1, \ldots, n\}$) are the singular values of a contextual state matrix. We average $H$ across all attention heads for a specific layer.

The singular value distribution directly determines compressibility. Given a singular value decomposition (SVD), low-rank compression removes small singular values, since each singular value indicates the importance of its corresponding directional component. When the singular value entropy is low (a concentrated spectrum), most information lies in a few dominant singular values, and truncation preserves nearly all content. When entropy is high (a flatter spectrum), removing any singular value entails noticeable information loss, making compression significantly harder.

**Receptive Field Analysis.** To further validate the effectiveness of SSE in long-context modeling, we analyze the receptive field of a pretrained 2B SSE model and compare it with that of a GLA baseline. As visualized in Figure 10, we compute the receptive field across all layers by examining the input-dependent gating matrices after pretraining (with a maximum sequence length of 8k). Specifically, we extract the last 128 tokens and measure the effective receptive field width across different channels in each layer using a threshold of 0.001. The results show that SSE consistently exhibits larger receptive fields than GLA across all layers, confirming SSE's enhanced capacity for long-range information integration.

## F.2 EXTENDED EXPERIMENTAL RESULTS

**12B SSE-H: Conversion and Reasoning Results.** To further evaluate the scalability of SSE, we convert a pretrained 12B Transformer into an SSE-H variant. The conversion involves a layer-wise

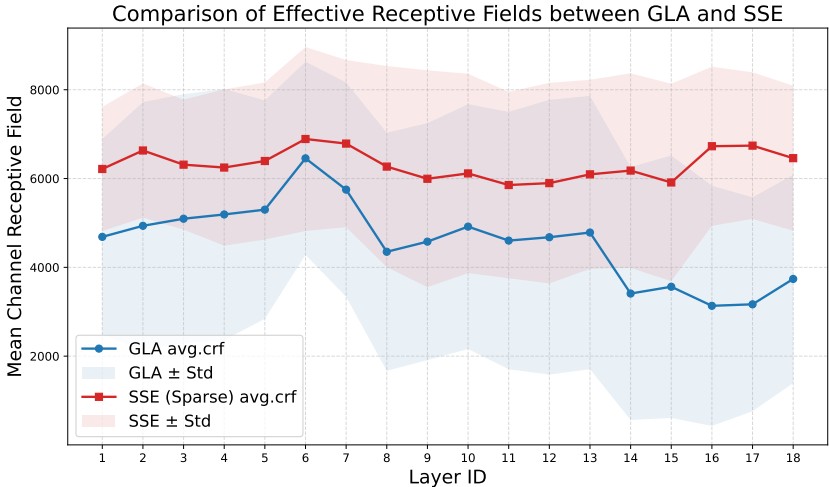

Figure 10: **SSE exhibits larger receptive fields than GLA.** We visualize the receptive field width of pretrained 2B SSE and GLA models after 8k-seqlen training. Values are computed over the last 128 tokens using a threshold of 0.001. SSE shows consistently broader receptive fields across all layers.

stacking pattern of SSE–SWA–SSE–SWA–Softmax layers, applied during the 128k long-context training stage, followed by supervised distillation. As shown in Table 9, SSE-H matches the mathematical reasoning accuracy of its softmax-attention counterpart, demonstrating the model's scalability to larger model sizes.

Table 9: **Reasoning ability of the 12B SSE-H variant under the conversion paradigm.**

| Model | AIME24 | AIME25 | MATH500 | OlympiadBench | AMC23 |
|---|---|---|---|---|---|
| Transformer-12B | 75.7 | 58.7 | 95.8 | 82.7 | 97.2 |
| **SSE-H-n4k1-12B** | 74.3 | 57.3 | 96.0 | 85.3 | 96.3 |

**2B SSE-H: Results on 32k RULER Benchmarks.** As discussed in Section 5.1, we scale the 2B SSE-H and Transformer baselines to 2T pretraining tokens, followed by a 32k extension with 250B additional tokens. We evaluate on six single- and multi-needle tasks from the RULER benchmark at lengths up to 32k. As shown in Table 10, SSE-H demonstrates strong long-context retrieval, matching or surpassing Transformer on several 32k tasks. These results underscore the robustness and effectiveness of our hybrid design under large-scale and long-context training.

Table 10: **Performance comparison on NIAH tasks in RULER after long-context extension.** All models have 2B parameters with a context length of 32k. Results are reported in a zero-shot setting.

| Model | S-NIAH-1 | | | S-NIAH-2 | | | S-NIAH-3 | | |
|---|---|---|---|---|---|---|---|---|---|
| | 8K | 16K | 32K | 8K | 16K | 32K | 8K | 16K | 32K |
| Transformer | 100.0 | 100.0 | 99.4 | 100.0 | 100.0 | 70.8 | 97.8 | 98.8 | 67.4 |
| **SSE-H-n4k1** | 100.0 | 100.0 | 93.6 | 100.0 | 100.0 | 84.4 | 100.0 | 98.2 | 90.2 |

| Model | MK-NIAH-1 | | | MQ-NIAH | | | MV-NIAH | | |
|---|---|---|---|---|---|---|---|---|---|
| | 8K | 16K | 32K | 8K | 16K | 32K | 8K | 16K | 32K |
| Transformer | 95.8 | 90.0 | 55.4 | 86.1 | 65.5 | 28.6 | 92.4 | 53.0 | 18.1 |
| **SSE-H-n4k1** | 91.0 | 84.8 | 64.6 | 89.1 | 67.8 | 44.6 | 87.3 | 66.8 | 40.6 |

**2B SSE-H: 32k Retrieval after Conversion.** As discussed in Section 5.1, we explore a conversion strategy that replaces a subset of softmax attention layers in a pretrained Transformer with linear

attention layers, followed by continued training on a limited dataset. Specifically, we apply this method to a 2B Transformer during its 128k long-context training stage, using approximately 100B tokens. We evaluate the resulting models on three 32k-context retrieval tasks. As shown in Table 11, the converted SSE-H model achieves substantially better retrieval than GLA-H and further narrows the performance gap with the original Transformer.

Table 11: **Evaluation of long-range retrieval (32k) under the conversion paradigm.**

| Model | Params. | Quest | Qampari | Table Query | Avg. |
|---|---|---|---|---|---|
| Transformer | 2B | 49.1 | 71.5 | 60.6 | 60.4 |
| GLA-H | 2B | 25.6 | 56.6 | 48.2 | 43.5 |
| **SSE-H-n4k1** | 2B | 35.6 | 70.6 | 50.0 | 52.1 |

**Full Ablation on Row Selection and Parameter Sharing.** As discussed in Section 5.5, we ablate softmax row selection and partition parameter sharing on 600M models with 15B tokens. Full results are provided in Tables 12 and 13, confirming the effectiveness of these design choices.

Table 12: **Full ablation on the effect of softmax-based row selection.**

| Models | Params. | Tokens | FDA | SWDE | SQuAD | Avg. |
|---|---|---|---|---|---|---|
| SSE-n4k1-k.silu | 600M | 15B | 20.87 | 26.91 | 24.90 | 24.23 |
| SSE-n4k1-k.softmax | 600M | 15B | 33.67 | 33.30 | 26.51 | 31.16 (+6.93) |

Table 13: **Full ablation on shared parameters.** SSE shares attention parameters across partitions, while "w/o shared-params" assigns separate parameters to each partition.

| Models | Params. (non-embed) | Tokens | Wiki.↓ | FDA | SWDE | SQuAD | Recall-Avg. | CSR-Avg. |
|---|---|---|---|---|---|---|---|---|
| SSE-n4k1 | 600M (300M) | 15B | 35.52 | 33.67 | 33.30 | 26.51 | 31.16 | 42.91 |
| w/o shared-params | 890M (580M) | 15B | 34.94 | 16.52 | 36.45 | 24.87 | 25.95 (-5.21) | 42.99 (+0.08) |

### F.3 EXPERIMENTAL RESULTS OF SSE-GDN

We primarily focus on SSE integrated with GLA-style transitions in the main experiments. Here, we provide a more comprehensive evaluation of integrating the delta-rule mechanism into SSE, denoted as SSE-GDN. Overall, SSE-GDN achieves state-of-the-art recall performance, and its hybrid variant (SSE-GDN-H) consistently outperforms GDN-H in hybrid architectures.

**600M SSE-GDN: Recall Performance.** Following the same training setting as in Table 1, SSE-GDN establishes a new state-of-the-art on recall benchmarks (Table 14). In addition, our write-read gating design remains effective compared with read-gate-only variants adopted in prior work such as MoM.

**15B MoE (700M Active) SSE-GDN-H: Scaling Performance.** Building upon the SSE-H results based on GLA, we further evaluate SSE-GDN-H. We use a 15B MoE model with 700M active parameters, trained on 450B tokens with a 1:6 hybrid ratio. The results are shown in Table 15. We observe that: (1) SSE-GDN-H achieves significant gains over both GDN-H and the Transformer baseline on MMLU and other general benchmarks; (2) it delivers substantial improvements on RULER, surpassing GDN-H and matching Transformer performance under the n4k2 setting, while maintaining strong extrapolation up to 16k tokens (with training context lengths $\leq$ 8k).

These results demonstrate that SSE extends effectively beyond diagonal linear attention and remains well-suited for delta-rule–based architectures.

### F.4 EFFICIENCY ANALYSIS

In Figure 4, we compare the speed (forward + backward time) of different SSE implementations. The masking implementation is competitive for short sequences ($\leq$1k), whereas the varlen implementation becomes more efficient as sequence length increases. Here, we further expand our ef-

Table 14: **Recall performance comparison on 600M models trained with 15B tokens.**

| Models | Params. | Tokens | FDA | SWDE | SQuAD | Avg. |
|---|---|---|---|---|---|---|
| Transformer | 600M | 15B | 74.50 | 60.67 | 32.67 | 55.95 |
| GLA | 600M | 15B | 9.44 | 23.40 | 23.06 | 18.63 |
| Mamba | 600M | 15B | 6.35 | 21.60 | 24.93 | 17.63 |
| Mamba2 | 600M | 15B | 20.51 | 30.42 | 27.21 | 26.05 |
| KDA | 600M | 15B | 24.50 | 35.55 | 26.71 | 28.92 |
| GDN | 600M | 15B | 14.79 | 33.48 | 26.24 | 24.84 |
| MoM-n4k1 | 800M (A600M) | 15B | 18.97 | 36.36 | 27.75 | 27.69 |
| MoM-n4k2 | 800M (A600M) | 15B | 21.05 | 37.08 | 26.68 | 28.27 |
| SSE-n4k1 | 600M | 15B | 33.67 | 33.30 | 26.51 | 31.16 |
| SSE-GDN-n4k1 (read gate) | 600M | 15B | 25.77 | 38.25 | 28.02 | 30.68 |
| **SSE-GDN-n4k1** | 600M | 15B | **41.56** | **41.13** | **30.83** | **37.84** |

Table 15: **SSE-GDN-H performance under the 15B MoE setting.**

| Model | MMLU | MMLU-Pro | C-Eval | AGIEval | BBH | DROP | GSM8k | TriviaQA | Ruler (4k) | Ruler (8k) | Ruler (16k) |
|---|---|---|---|---|---|---|---|---|---|---|---|
| Transformer | 61.5 | 60.6 | 64.6 | 62.3 | **80.0** | 67.5 | 85.1 | 73.0 | **89.5** | **81.5** | 0.0 |
| GDN-H (1:6) | 62.1 | 62.4 | 64.6 | 60.9 | 77.1 | 68.5 | 86.3 | 72.4 | 81.0 | 63.6 | 38.9 |
| **SSE-GDN-H-n4k1 (1:6)** | 62.1 | 61.8 | 65.3 | 61.5 | 79.4 | **71.1** | 86.9 | 71.3 | 82.0 | 72.1 | 46.6 |
| **SSE-GDN-H-n4k2 (1:6)** | **64.7** | **64.7** | **67.7** | **64.3** | 78.3 | 69.1 | **87.6** | **73.2** | 88.5 | 81.2 | **48.7** |

ficiency study through two complementary evaluations: (1) attention runtime and (2) end-to-end training throughput.

**Attention Runtime (Forward + Backward, ms).** We benchmark SSE (with GLA transition), GLA, and full attention across sequence lengths ranging from 8k to 500k. As shown in Table 16, SSE demonstrates linear scaling behavior and becomes substantially faster than full attention beyond 32k tokens. Compared with GLA, SSE incurs moderate overhead due to router index sorting, Q/K/V/O reordering, increased varlen segmentation (resulting in more kernel-level chunk boundaries), and computing attention over $k+1$ state partitions per step.

**Training Throughput (K tokens/s).** We measure training throughput on a single A100 GPU. For sequence lengths from 2k to 16k, we use a 1.3B model; for 32k, we use a 340M model due to memory constraints. We additionally evaluate SSE-GDN, which integrates SSE with the GDN architecture. Aggregated results are summarized in Table 17.

Overall, SSE surpasses Transformer in throughput at 32k sequence length, while remaining slower than GLA, GDN, and Mamba2 due to the inherent overhead introduced by state expansion, reflecting a clear speed-performance tradeoff. SSE-GDN achieves approximately 70% of GDN throughput (comparable to Mamba), and SSE(-GLA) reaches approximately 60% of GLA throughput. Despite this gap, SSE maintains sufficient efficiency for large-scale long-context pretraining. We consider targeted low-level kernel optimization a promising direction for further reducing the remaining overhead.

Table 16: **Attention runtime comparison across sequence lengths.** Runtime is measured as the time (ms) required for a forward and backward pass.

| Model / Seq Len | 8k | 16k | 32k | 64k | 128k | 256k | 500k |
|---|---|---|---|---|---|---|---|
| Full Attention (Quadratic) | 6 | 9 | 24 | 83 | 315 | 1250 | 4376 |
| GLA (Linear) | 3 | 5 | 10 | 18 | 36 | 71 | 131 |
| **SSE-n4k1-varlen** (Linear) | 8 | 14 | 26 | 50 | 97 | 191 | 358 |

Table 17: **Training throughput comparison.** Throughput is measured in K tokens/s on a single A100 GPU. Models with 1.3B parameters are used for sequence lengths from 2k to 16k, while a 340M model is used for 32k due to memory constraints.

| Model / Seq Len & Batch Size | 2k & 8 | 4k & 4 | 8k & 2 | 16k & 1 | 32k & 1 |
|---|---|---|---|---|---|
| Transformer | 24.24 | 22.35 | 19.41 | 15.25 | 25.14 |
| GLA | 19.55 | 19.47 | 19.44 | 19.23 | 53.01 |
| GDN | 21.37 | 21.07 | 20.87 | 20.48 | 55.71 |
| Mamba | 12.81 | 12.79 | 12.48 | 14.72 | 35.87 |
| Mamba2 | 18.32 | 18.37 | 18.13 | 18.29 | 45.81 |
| MoM-GDN-n4k1 | 11.12 | 11.23 | 11.16 | 10.20 | 23.71 |
| **SSE(-GLA)-n4k1** | 12.40 | 12.33 | 12.29 | 12.29 | 28.79 |
| **SSE-GDN-n4k1** | 14.88 | 14.86 | 14.73 | 14.45 | 35.08 |

