# OpenReview forum: "Scaling Linear Attention Capacity with Sparse State Expansion"
_ICLR.cc/2026/Conference — ICLR 2026 Poster_

### Official Review · Reviewer_KhC1 · 2025-10-21

**Soundness:** 3
**Presentation:** 3
**Contribution:** 3
**Rating:** 6
**Confidence:** 4

**Summary:**

This paper proposes a new framework to enhance the efficiency and scalability of linear attention models for long-context processing. It introduces two innovations: (1) row-sparse state updates, which treat information storage as a classification task using top-k softmax selection to reduce interference and extend context range, and (2) Sparse State Expansion (SSE), which partitions the state into multiple sparsely updated segments to increase memory capacity without adding parameters. Experiments show that SSE and its hybrid variant (SSE-H) outperform earlier linear attentions across language modeling, retrieval, and reasoning tasks.

**Strengths:**

Clear motivation, sound implementation, and good performance with model at scale.

**Weaknesses:**

1. From the Fig.4 we can see that the wall time is not linear with respect to the sequence length. Why? Hope the authors could provide some more experiment results comparing the wall time of their architecture against earlier linear attentions, GLA, GDN, Mamba2, and full attention.
2. Hope the authors could also provide scaling experiments compare their architecture against earlier linear attentions, GLA, GDN, Mamba2, and full attention, considering training wall time vs perplexity.
3. Hope the authors could also discuss MoE for quadratic attention in their related work.

**Questions:**

On which datasets was the model trained? Will these data be open-sourced?

---

> ### Author Response · Authors · 2025-11-19
> **[1/3] Response to Reviewer KhC1 (W1)**
>
> > **W1: Why does Fig. 4 not show linear wall-clock scaling, and can the authors provide additional runtime comparisons?**
>
> Thank you for the question. In Fig. 4, **the x-axis is not uniformly spaced** (the sequence length increases by powers of two) while the y-axis uses a uniform scale. This makes the curve appear non-linear even though the underlying complexity is linear. We apologize for the confusion and will consider using a log2 scale for clearer visualization in a future revision.
>
> Regarding more comprehensive efficiency comparisons：
>
> 1. **Attention runtime (forward + backward, ms)**.
> We benchmarked SSE, GLA, and full attention from 8k to 500k sequence lengths. **SSE follows linear scaling and becomes significantly faster than full attention beyond 32k**, while being moderately slower than GLA due to overheads such as sorting router indices, reordering Q/K/V/O, increased varlen segments (leading to more kernel-level chunk boundaries), and computing attention over $k{+}1$ state partitions per step.
>
> | Model / Seq Len            |  8k  | 16k | 32k | 64k | 128k | 256k  | 500k |
> |------------------|-------|--------|-------|-------|-------|-------|--------------|
> | Full Attention (Quadratic)   | 6 | 9  | 24 | 83 | 315 | 1250 | 4376 |
> | GLA (Linear)                 | 3 | 5  | 10 | 18 | 36 | 71 | 131|
> | SSE-n4k1-varlen (Linear)     | 8 | 14  | 26 | 50 | 97 | 191 | 358 |
>
> 2. **Training throughput (K tokens/s).**
> We further measured throughput on a single A100, using a 1.3B model for 2k–16k sequences and a 340M model for 32k.
> We also included SSE-GDN (SSE combined with GDN); see Reviewer Jg4D Weakness 1 for details.
> The aggregated results are shown below.
>
> | Model / Seq Len & Batch Size            | 2k & 8 | 4k & 4  | 8k & 2 | 16k & 1 | 32k & 1 (340M params)|
> |------------------|-------|-------|--------|-------|-------|
> | Transformer++                | 24.24  | 22.35  | 19.41  | 15.25   | 25.14 |
> | GLA                          | 19.55  | 19.47  | 19.44  | 19.23   | 53.01 |
> | GDN                          | 21.37  | 21.07  | 20.87  | 20.48   | 55.71 |
> | Mamba                        | 12.81  | 12.79  | 12.48  | 14.72   | 35.87 |
> | Mamba2                       | 18.32  | 18.37  | 18.13  | 18.29   | 45.81 |
> | MoM-GDN-n4k1                 | 11.12  | 11.23  | 11.16  | 10.20   | 23.71 |
> | **SSE-GLA-n4k1**             | 12.40  | 12.33  | 12.29  | 12.29   | 28.79 |
> | **SSE-GDN-n4k1**             | 14.88  | 14.86  | 14.73  | 14.45   | 35.08 |
>
>
> Overall, SSE is faster than Transformer++ at 32k, but is slower than GLA/GDN/Mamba2 because of the overhead inherent to state expansion, **reflecting a clear speed–performance tradeoff**. Even so, SSE still provides sufficiently high training efficiency to support large-scale long-context training, and we view targeted low-level optimization as a promising direction for further reduce this gap.
>
> ---

---

> ### Author Response · Authors · 2025-11-19
> **[2/3] Response to Reviewer KhC1 (W2)**
>
> > **W2: Scaling experiments comparing SSE with earlier linear-attention baselines and full attention, evaluating training time versus perplexity.**
>
> Thank you for raising this point.
>
> 1. **Scaling experiments:**
> We provide 2B-parameter scaling results trained on 20B/40B/100B tokens using a constant learning rate. As shown in the table, SSE scales effectively: **as the number of training tokens increases, SSE gradually achieves the best performance across all metrics**, while maintaining a stable advantage on recall at all scales.
> The reported metrics includes Wikitext and LAMBADA(-Standard/OpenAI) perplexity, CSR-Avg over six commonsense reasoning tasks, and Recall-Avg over three real-world recall tasks. The experiments cover both the original SSE-GLA and the newly added SSE-GDN (see our response to Reviewer Jg4D Weakness 1 for details).
>
> 2. **Training wall time:**
> Regarding training efficiency, SSE provides sufficiently high throughput to support large-scale long-context training, though a speed gap versus earlier linear-attention variants remains due to state expansion.
> Because training wall time is highly dependent on specific training setups (e.g., hardware scale, parallelism strategy, and sequence length), a universal wall-time–vs–PPL scaling curve is difficult to report. Instead, we include **8k-sequence throughput as a reference metric**, and more comprehensive throughput results are provided in our response to Weakness 1. Note that linear attention models typically surpass Transformers in throughput primarily at longer sequence lengths.
>
> | Model        | Tokens | Throughput (8k) | Wiki. (PPL) | Lamb-S. (PPL) | Lamb-O. (PPL) | CSR-Avg. | Recall-Avg. |
> | ------------ | ------ | ---------- | ---------- | ------------ | ------------ | ------- | ---------- |
> | Transformer  | 20B    | 19.41      | 24.42      | 27.64        | 14.94        | 46.16   | 64.62      |
> | GLA          | 20B    | 19.44      | 33.88      | 71.67        | 29.49        | 43.48   | 27.04      |
> | Mamba2       | 20B    | 18.13      | 26.50      | 28.19        | 17.85        | 46.56   | 32.34      |
> | GDN          | 20B    | 20.87      | 25.51      | 27.03        | 15.32        | 46.91   | 34.94      |
> | SSE-GLA-n4k1 (Ours) | 20B    | 12.29      | 25.94      | 28.56        | 17.71        | 46.80   | 38.62      |
> | SSE-GDN-n4k1 (Ours) | 20B    | 14.73      | 24.81      | 29.33        | 16.43        | 46.98   | 45.08      |
> | - | -| - | - | - | - | - | - |
> | Transformer  | 40B    | 19.41      | 20.36      | 14.91        | 10.87        | 49.71   | 68.20      |
> | GLA          | 40B    | 19.44      | 27.47      | 39.57        | 20.31        | 45.29   | 35.47      |
> | Mamba2       | 40B    | 18.13      | 21.71      | 18.08        | 13.44        | 50.05   | 42.10      |
> | GDN          | 40B    | 20.87      | 21.28      | 15.26        | 10.50        | 49.42   | 45.84      |
> | SSE-GLA-n4k1 (Ours) | 40B    | 12.29      | 21.37      | 13.38        | 11.15        | 50.55   | 49.03      |
> | SSE-GDN-n4k1 (Ours) | 40B    | 14.73      | 20.84      | 17.22        | 12.05        | 49.61   | 53.31      |
> | - | -| - | - | - | - | - | - |
> | Transformer  | 100B   | 19.41      | 17.76      | 10.89        | 7.87         | 52.23     | 72.73      |
> | GLA          | 100B   | 19.44      | 23.03      | 16.35        | 11.35        | 48.39     | 43.29      |
> | Mamba2       | 100B   | 18.13      | 18.98      | 12.23        | 8.51         | 52.34     | 48.47      |
> | GDN          | 100B   | 20.87      | 18.49      | 10.82        | **7.85**     | 53.09     | 48.71      |
> | SSE-GLA-n4k1 (Ours) | 100B   | 12.29      | **18.44**  | **10.68**    | 8.04         | **53.63** | **56.63**  |
> | SSE-GDN-n4k1 (Ours) | 100B   | 14.73      | **18.20**  | **10.22**    | **7.85**     | **53.14** | **60.45**  |
>
> ---

---

> ### Author Response · Authors · 2025-11-19
> **[3/3] Response to Reviewer KhC1 (W3 and Q1)**
>
> > **W3: The authors should also discuss MoE methods for quadratic attention in the related-work section.**
>
> Thank you for the suggestion. We will incorporate a discussion of MoE approaches applied to quadratic attention into the related-work section. These include MoE on attention projections (e.g., Switch Transformers [1], SwitchHead [2]) as well as MoE at the head level (e.g., MoH [3], MoSA [4], MOA [5]). We will ensure these connections are clearly articulated in the revised version.
>
> [1] Switch Transformers: Scaling to Trillion Parameter Models with Simple and Efficient Sparsity. https://arxiv.org/abs/2101.03961
> [2] SwitchHead: Accelerating Transformers with Mixture-of-Experts Attention. https://arxiv.org/abs/2312.07987
> [3] MoH: Multi-Head Attention as Mixture-of-Head Attention. https://arxiv.org/abs/2410.11842
> [4] Mixture of Sparse Attention: Content-Based Learnable Sparse Attention via Expert-Choice Routing. https://arxiv.org/abs/2505.00315
> [5] MoA: Mixture of Sparse Attention for Automatic Large Language Model Compression. https://arxiv.org/abs/2406.14909
>
> ---
>
> > **Q1: On which datasets was the model trained? Will these data be open-sourced?**
>
> The model is trained on a mixture of privately held text corpora. Due to data-usage agreements, the full training corpus cannot be released. We will provide additional details on the data composition in the final (non-anonymous) version.
>
> ---

---

> ### Comment · Reviewer_KhC1 · 2025-11-22
> **Thank the authors for their rebuttal.**
>
> Since all my concerns are well-addressed, I raise my score to 8.

---

> > ### Author Response · Authors · 2025-11-22
> >
> > Thank you very much for your thoughtful feedback and for raising your score. We sincerely appreciate your careful evaluation and are glad that our responses addressed your concerns. Your comments have been invaluable in improving the clarity and quality of our work.

---

### Official Review · Reviewer_heMc · 2025-10-26

**Soundness:** 4
**Presentation:** 2
**Contribution:** 4
**Rating:** 8
**Confidence:** 3

**Summary:**

This paper improves context compression in linear attention models with row-sparse update and sparse state expansion (SSE). The row-sparse update learns a sparse mask with top-k and softmax to update only a few rows in the contextual states, which utilizes the state space more effectively than dense update. Since different tokens are associated with different rows, the row-sparse update eliminates the need of gating at each step, thereby avoiding the limited receptive field caused by gating. The authors then propose SSE to extend row-sparse update to larger state space. SSE first divides a large state space into N partitions, and then perform dense updates on k partitions. By sharing the attention parameters across partitions, SSE decouples the parameters size and the memory capacity, thereby solving the capacity bottleneck of linear attention models. Experiments on language modeling, needle-in-a-haystack and reasoning benchmarks show that SSE and the hybrid SSE-H achieves state-of-the-art performance among linear attention models.

**Strengths:**

1. This paper tackles the memory capacity problem, a key issue in linear attention models with clear motivations. The row-wise sparse update aims to improve the utilization of a fixed size state space, while the SSE extends the memory capacity without increasing the number of parameters.
2. Experiments are thorough and solid. The authors visualizes the cosine similarity of the state space and show that row-sparse update significantly improves state space utilization compared to existing designs. The final SSE model is evaluated against Transformer and linear attention baselines on a wide range of benchmarks, with a rigorous setup of fixed number of parameters.
3. The efficiency of SSE is not confined to theory. The authors implemented SSE by grouping tokens into subsequences according to their partitions and executing them with a linear attention kernel call. This makes SSE practical for real-world long context use.

**Weaknesses:**

1. The writing of this paper may be largely improved. The authors used quite a few terminologies or preliminaries without enough explanations, which make this paper hard to follow. For example, the authors didn’t explain how they computed the cosine similarity in Figure 1. Line 176 compares SSE against gated linear attention, but the form of gated linear attention is never mentioned in the paper. There aren’t ground truth classes nor a classification task, but the authors keep using the term classification to refer to state space utilization. In Line 245, the term segmented clustering is used, but this isn’t a commonsense for audience. Captions of figure 1 and 3 need to be extended with their implications. See more in questions.
2. The title doesn’t exactly reflect the contribution of this paper. “Scaling linear attention" sounds like this paper studies the model performance under different linear attention sizes. I would suggest modifying it to be “Extending linear attention capacity with sparse state expansion”.
3. While SSE focuses on improving linear attention models, there isn’t any comparison of wall time for SSE and baselines. Could you please report the performance-time trade-off curve for SSE, Transformer and other linear attention models? That will help audience know which model to use given a specific context length.

**Questions:**

1. Is the cosine similarity computed for a single step then averaged over a sequence or the whole dataset?
2. The logic in Line 174-178 is hard to understand. It’s hard to understand what Propositions 2-4 are without looking into the Appendix. Propositions 2-4 use the row-sparse update, which is introduced only in later sections. Besides, it’s not very clear how the conclusion of decay in gated variants is derived before looking into Proposition 4. As row-sparse update is complementary to gating and they may co-exist, it’s also hard to understand why SSE solves the issue caused by gating. You may discuss eliminating gating as a benefit of SSE after introducing the method.
3. Line 184-186: By theoretical analysis, do you mean Proposition 2? It’s a little bit hard to understand this without looking into the appendix. You need to add more details in the main paper.
4. The functions softmax and top-k produce a $k$ dimension vector by their definition, which is not correct. Do you mean softmax on the $k$ non-zero elements and keep the rest as 0? Then you need to re-define your softmax function.
5. Line 263-265: This key insight should be brought to early paragraphs of Sec 4.1. Otherwise, it’s hard to understand how SSE is connected with row-sparse update. You’re essentially factorizing a row-sparse update for N*c rows into two small parameter matrices.
6. Font size in Figure 3 & 4 should be increased.
7. Line 284 & 286: Why can the always-selected partition capture local interactions? The inputs are accumulated by addition, which doesn’t model interactions.
8. Line 294-296: Please explain why singular value entropy reflects the difficulty of compression. It’s not a commonsense to audience.
9. Line 301: Sequential computation is never explained in the main paper.
10. Line 308-310: I would recommend to draw a figure for grouping and varlen technique.
11. In Table 1, linear attention models are worse than Transformer. Is it because language modeling requires more pairwise interactions? Then why SSE becomes on par with Transformer in Table 3 & 4?
12. Figure 5: Which task is it?
13. Figure 6: Does SSE-Shared refer to the always-selected partition? Please be consistent in the terms.
14. Line 457: Is the sparsity measured within the selected partitions? For n4k1, I understand the upper bound of sparsity for all partitions should be 25%, right?

---

> ### Author Response · Authors · 2025-11-19
> **[1/3] Response to Reviewer heMc (W1~W3)**
>
> > **W1: The writing may be improved; several terms and preliminaries lack clear explanations.**
>
> Thank you for the detailed feedback. We will revise the manuscript to clarify all missing definitions and strengthen exposition. Below we address each specific point.
>
> ---
>
> > **W1.1: Cosine similarity in Figure 1 is not explained.**
>
> We will add a clear description of how cosine similarity is computed and include this explanation directly in the caption (also detailed in Question 1).
>
> ---
>
> > **W1.2: Gated linear attention is compared but never defined.**
>
> We will introduce the formulation of GLA in the main text before any comparison (see Question 2).
>
> ---
>
> > **W1.3: Use of the term "classification" is confusing.**
>
> We will refine this terminology. Our intention was not to imply a supervised task, but to express that different state rows act as latent categories, and that $f(\mathbf{x}_t, \mathbf{W}_k)$ produces soft assignment scores used to select which rows to update. This motivates the Softmax-top-$k$ mechanism, but we agree that calling this "classification" can be misleading and will refine the terminology accordingly.
>
> ---
>
> > **W1.4: "Segmented clustering" is unclear.**
>
> We will replace this term. Our intent was to describe how SSE, through partition-wise parameter sharing, induces a re-segmentation of the sequence where each segment corresponds to a different state partition. Since "segmented clustering" is not standard and may confuse readers, we will adopt clearer wording in the revision.
>
> ---
>
> > **W1.5: Captions of Figures 1 and 3 are too short.**
>
> We will extend both captions to include their implications, incorporating clarifications from Questions 1 and 8.
>
> ---
>
> > **W2: The title does not accurately reflect the contribution of the paper.**
>
> Thank you for the helpful suggestion. We will revise the title to more clearly express the contribution.
>
> Our original intention in using "scaling" was to emphasize how performance improves as the state size increases (as shown in Figure 5), rather than to suggest a study of model-size scaling. We agree that a clearer phrasing would better reflect the method's purpose and will consider alternatives such as **Extending/Scaling Linear Attention Capacity with Sparse State Expansion** in the revised version.
>
> ---
>
> > **W3: No wall-time or performance–time trade-off comparison is provided for SSE.**
>
> Thank you for raising this point. We have added runtime comparisons to clarify the performance–efficiency trade-off of SSE.
>
> 1. We benchmarked **attention runtime (forward + backward, in ms)** for SSE, GLA, and full attention over sequence lengths from 8k to 500k.
> SSE scales favorably: **it becomes substantially faster than full attention at long sequence lengths (>32k)**, while being moderately slower than GLA due to additional operations such as sorting router indices, reordering Q/K/V/O, and computing attention over $k{+}1$ state partitions per step.
>
> | Model / Seq Len            |  8k  | 16k | 32k | 64k | 128k | 256k  | 500k |
> |------------------|-------|--------|-------|-------|-------|-------|--------------|
> | Full Attention (Quadratic)   | 6 | 9  | 24 | 83 | 315 | 1250 | 4376 |
> | GLA (Linear)                 | 3 | 5  | 10 | 18 | 36 | 71 | 131|
> | SSE-n4k1-varlen (Linear)     | 8 | 14  | 26 | 50 | 97 | 191 | 358 |
>
> 2. We measured **training throughput (K tokens / s)** on a single A100 GPU, using a 1.3B model for 2k–16k sequences and a 340M model for 32k.
> We also included SSE-GDN (SSE combined with GDN; see our response to Reviewer Jg4D Weakness1 for details).
> Below is the summarized throughput table:
>
> | Model / Seq Len & Batch Size            | 2k & 8 | 4k & 4  | 8k & 2 | 16k & 1 | 32k & 1 (340M params)|
> |------------------|-------|-------|--------|-------|-------|
> | Transformer++                | 24.24  | 22.35  | 19.41  | 15.25   | 25.14 |
> | GLA                          | 19.55  | 19.47  | 19.44  | 19.23   | 53.01 |
> | GDN                          | 21.37  | 21.07  | 20.87  | 20.48   | 55.71 |
> | Mamba                        | 12.81  | 12.79  | 12.48  | 14.72   | 35.87 |
> | Mamba2                       | 18.32  | 18.37  | 18.13  | 18.29   | 45.81 |
> | MoM-GDN-n4k1                 | 11.12  | 11.23  | 11.16  | 10.20   | 23.71 |
> | **SSE-GLA-n4k1**             | 12.40  | 12.33  | 12.29  | 12.29   | 28.79 |
> | **SSE-GDN-n4k1**             | 14.88  | 14.86  | 14.73  | 14.45   | 35.08 |
>
>
> Overall, SSE is faster than Transformer++ at 32k, but remains slower than GDN and GLA because of the overhead inherent to state expansion, **reflecting a clear speed–performance tradeoff**. Nevertheless, SSE still provides sufficiently high training efficiency to support large-scale long-context training, and we view targeted low-level optimization as a promising next step for further improving throughput.
>
> ---

---

> ### Author Response · Authors · 2025-11-19
> **[2/3] Response to Reviewer heMc (Q1~Q7)**
>
> > **Q1: Is the cosine similarity computed for a single step then averaged over a sequence or the whole dataset?**
>
> In Figure 1, cosine similarity is computed on the *final state* of each sequence rather than at every step. This avoids unnecessary overhead and yields a more stable signal for evaluating row-wise separation. For the reported results, we sample multiple long-context examples from our benchmarks (Table 11) and average the cosine similarities across these sequences. Using long inputs ensures that the state values have sufficiently converged, and we observe consistent similarity patterns across all sampled cases.
>
> ---
>
> > **Q2: The logic in Lines 174–178 is hard to follow, and the connection to Propositions 2–4 and the row-sparse update is unclear.**
>
> Thank you for the helpful feedback.
>
> (1) Due to space constraints, we placed Propositions 2–4 in the Appendix.C and presented only the key conclusions in the main text. In future revisions, we will include simplified propositions directly in the main paper and relocate the discussion in Lines 174–178 to after the introduction of the row-sparse update in Section 3.2.
>
> (2) The row-sparse state update in our method contains two components—decay of historical information and addition of current information (only the state rows corresponding to the $k$ largest values in $\mathbf{k}_t$ are updated at each step). We will explicitly mark the decay term in Eq. 6 and discuss the benefit of eliminating gating to make this clearer and avoid confusion. In addition, we will move Proposition 4 into the main text in the final version.
>
> ---
>
> > **Q3: The reference to "theoretical analysis" in Lines 184–186 is unclear. Does it refer to Proposition 2? More detail is needed in the main paper.**
>
> The "theoretical analysis" refers to *Propositions 2 and 3*, which show that hard classification causes each state row to store information associated with the same latent category, leading to a more structured organization of the state space. We will make this explicit in the main text and provide sufficient detail so that readers do not need to rely on the appendix to understand the argument.
>
> ---
>
> > **Q4: The description of softmax and top-$k$ is unclear. A d-dimensional vector is mentioned, which seems incorrect. Do you mean applying softmax only on the top-$k$ non-zero elements and keeping the rest as 0? If so, this needs to be re-defined.**
>
> Thank you for pointing this out. Your understanding is correct: we define the operation $\operatorname{softmax}(\operatorname{top\text{-}}k(\mathbf{x}))$ as applying softmax only to the top-$k$ selected entries of $\mathbf{x}$, while setting all other entries to zero. In the revision, we will explicitly state this definition in the main text to avoid ambiguity.
>
> ---
>
> > **Q5: The key insight in Lines 263–265 should appear earlier in Sec. 4.1; otherwise the connection between SSE and the row-sparse update is unclear.**
>
> Thank you for the suggestion. This insight arises from the combination of softmax row selection and shared attention parameters. We will move this explanation to the opening (two) paragraphs of Sec. 4.1 in the revised version to make the connection clearer.
>
> ---
>
> > **Q6: The font size in Figures 3 and 4 is too small.**
>
> Thank you for the suggestion. We will increase the font size in Figures 3 and 4 in the revised version to ensure better readability.
>
> ---
>
> > **Q7: Why can the always-selected partition capture local interactions? Addition-based accumulation does not seem to model interactions.**
>
> In linear attention, the **recurrent state is the only channel through which historical information influences future tokens**. Thus, state recurrence itself encodes token–token interactions. From the attention-score view,
> $$
> \mathbf{q}_t\mathbf{S}_t = \mathbf{q}_t\left(\sum_i \mathbf{k}_i^\top \mathbf{v}_i\right)
> = \sum_i (\mathbf{q}_t\mathbf{k}_i^\top)\,\mathbf{v}_i,
> $$
> showing that interactions arise whenever $\mathbf{q}_t\mathbf{k}_i^\top \neq 0$.
>
> Under SSE, a sparse update imply that a partition receives no new input ($\mathbf{k}_t=\mathbf{0}$), causing its attention scores for future tokens to vanish. **Interactions therefore only persist within tokens that fall into the same partition**. However, adjacent tokens typically require dense, continuous interactions—similar to convolutions or sliding-window attention—and we cannot guarantee that they are always assigned to the same sparse partition.
>
> To address this, we introduce an always-selected partition. Combined with the decay mechanism, this partition **retains short-range information and continuously supplies local interaction flow**, while the remaining sparse partitions focus on selective, non-local updates.
>
> ---

---

> ### Author Response · Authors · 2025-11-19
> **[3/3] Response to Reviewer heMc (Q8~Q14)**
>
> > **Q8: Please explain why singular value entropy reflects the difficulty of compression.**
>
> The singular value distribution directly determines compressibility. Given an SVD $ \mathbf{U}\mathbf{S}\mathbf{V}^\top $, low-rank compression removes small singular values, since each singular value indicates the importance of its corresponding directional component. When the singular value entropy is low (a concentrated spectrum), most information lies in a few dominant singular values, and truncation preserves nearly all content. When entropy is high (a flatter spectrum), removing any singular value entails noticeable information loss, making compression significantly harder. We will add this clarification after introducing singular value entropy in the revised version.
>
> ---
>
> > **Q9: Sequential computation is never explained in the main paper.**
>
> Thank you for pointing this out. We only described sequential computation in Appendix D. In the revised version, we will add a clear explanation in the main text: sequential computation refers to invoking the linear-attention kernel separately for each partition in a *for-loop*, which avoids variable-length control and provides a simple though less efficient way to process all partitions in order.
>
> ---
>
> > **Q10: A figure should be added to illustrate the grouping and varlen techniques.**
>
> Thank you for the suggestion. We will add an illustrative figure in the revised version to clearly demonstrate how grouping and varlen techniques operate.
>
> ---
>
> > **Q11: In Table 1, linear attention models are worse than Transformer. Is this due to the need for more pairwise interactions in language modeling? Then why does SSE become comparable to Transformer in Tables 3 and 4?**
>
> Table 1 reports both language modeling (LM) benchmarks and recall-intensive tasks. Linear-attention models are not weaker than Transformers on LM-style evaluations, which do not heavily rely on precise long-range pairwise interactions. The performance gap appears only on the recall-focused tasks (FDA, SWDE, SQuAD), which require patterns such as "A, B, …, A → B", where accurate pairwise information flow is essential.
>
> In Table 3, most benchmarks (e.g., MMLU) are LM-style tasks, with only a few recall-heavy components (e.g., SWDE). Because SSE-H incorporates a hybrid architecture with full attention, the performance gap is small—consistent with the SSE-H results observed in Table 1.
>
> In Table 4, all evaluations involve mathematical reasoning. Although CoT traces may be long, our results indicate that these tasks demand only moderate pairwise interaction. SSE-H is sufficient to capture this level of dependency, which explains why it performs on par with Transformers.
>
> ---
>
> > **Q12: Which task is shown in Figure 5?**
>
> Figure 5 reports the average recall accuracy over three real-world recall tasks: FDA, SWDE, and SQuAD. Thank you for pointing this out; we will add this clarification in the revised version.
>
> ---
>
> > **Q13: Does SSE-Shared in Figure 6 refer to the always-selected partition? Please use consistent terminology.**
>
> Thank you for noticing this. Yes, SSE-Shared corresponds to the always-selected partition. We will revise the terminology in Figure 6 and use "always-selected partition" consistently throughout the paper.
>
> ---
>
> > **Q14: Is the sparsity measured within the selected partitions? For n4k1, the upper bound over all partitions should be 25%, correct?**
>
> Your understanding is correct: the reported sparsity is measured within the selected partitions. If sparsity is computed over all partitions, the value should indeed be scaled by 25% under the n4k1 setting (or by $2/5$ if the always-selected partition is counted).
>
> ---

---

> > ### Comment · Reviewer_heMc · 2025-11-24
> >
> > Thanks the authors for their proper response. It's good to learn that the speed of SSE surpasses Transformer when there are 64k or more tokens. Since my score was already positive, I decide to keep my score.

---

> > > ### Author Response · Authors · 2025-11-25
> > >
> > > Thank you very much for your encouraging response and for highlighting the points that strengthened our work. We have incorporated your suggestions into the revised PDF.

---

### Official Review · Reviewer_Jg4D · 2025-10-31

**Soundness:** 2
**Presentation:** 4
**Contribution:** 2
**Rating:** 4
**Confidence:** 4

**Summary:**

The paper proposes Sparse State Expansion (SSE) for linear attention. Two ideas drive the method: (i) row‑sparse updates that treat state updates as a classification problem and write only to top‑k rows via a softmax head, and (ii) state expansion into N shared‑parameter partitions chosen by a write–read gate, so capacity (state size) scales without growing parameter count. Efficient masked/varlen implementations are provided. Empirically, SSE and its hybrid variant are tested on language modeling, retrieval, and math reasoning.

**Strengths:**

1. Modeling state updates as information classification is well argued and operationalized
2. Decoupling the model’s parameters and state's parameters is important direction for improving linear transformers.

**Weaknesses:**

1. There is no convincing explanation on why SSE is only effective for linear attention but not deltanet.
2. Baselines seem to be cherry picked since they are neither the most powerful nor fundamentally relevant sequence models.
3. The paper uses hard partition selection (top‑k) and softmax row selection, but the gradient treatment for the discrete top‑k isn’t described. Can you please clarify how gradients flow through Eqs. 7–9 and whether any implementation tricks are needed for stability.

**Questions:**

See weaknesses.

---

> ### Author Response · Authors · 2025-11-19
> **[1/3] Response to Reviewer Jg4D (W1)**
>
> > **W1: There is no convincing explanation on why SSE is only effective for linear attention but not DeltaNet.**
>
> Thank you for raising this point. In the revision, we clarify that SSE is **not restricted to diagonal linear attention**, and we provide new experimental evidence demonstrating its effectiveness when combined with Gated-DeltaNet (denoted as SSE-GDN). Overall, **SSE-GDN achieves state-of-the-art recall, and SSE-GDN-H consistently outperforms GDN-H in hybrid architectures.**
>
> At the time of submission, our evaluation of SSE with GDN was incomplete—we had primarily focused on GLA—and we apologize for the confusion. Since then, we have conducted a more thorough investigation of integrating the delta-rule mechanism with SSE:
>
> - **Exp1 (600M params, 15B tokens): SSE-GDN establishes a new state-of-the-art on recall benchmarks.**
> In addition, our write–read gating design remains effective compared with the read-gate-only variants used in other works such as MoM. (Additional baselines are included; please see Weakness 2.)
>
> | Model                        | Size                | Tokens | FDA   | SWDE  | SQuAD | Avg.   |
> |-----------------------------|---------------------|--------|-------|-------|-------|-------|
> | Transformer                 | 600M                | 15B    | 74.50 | 60.67 | 32.67 | 55.95 |
> | GLA                         | 600M                | 15B    | 9.44  | 23.40 | 23.06 | 18.63 |
> | Mamba                       | 600M                | 15B    | 6.35  | 21.60 | 24.93 | 17.63 |
> | Mamba2                      | 600M                | 15B    | 20.51 | 30.42 | 27.21 | 26.05 |
> | KDA                         | 600M                | 15B    | 24.50 | 35.55 | 26.71 | 28.92 |
> | GDN                         | 600M                | 15B    | 14.79 | 33.48 | 26.24 | 24.84 |
> | MoM-n4k1                    | 800M (600M active)  | 15B    | 18.97 | 36.36 | 27.75 | 27.69 |
> | MoM-n4k2                    | 800M (600M active)  | 15B    | 21.05 | 37.08 | 26.68 | 28.27 |
> | SSE-GLA-n4k1                | 600M                | 15B    | 33.67 | 33.30 | 26.51 | 31.16 |
> | SSE-GDN-n4k1 (read gate)    | 600M                | 15B    | 25.77 | 38.25 | 28.02 | 30.68 |
> | **SSE-GDN-n4k1**                | 600M                | 15B    | **41.56** | **41.13** | **30.83** | **37.84** |
>
> - **Exp2 (15B MoE with 700M active, 450B tokens, 1:6 hybrid):**
> Extending the SSE-H results based on GLA, we further evaluate SSE-GDN-H and observe:
>     - **Significant gains over both GDN-H and the Transformer baseline on MMLU and other general benchmarks.**
>     -  **Substantial improvements on Ruler**, surpassing GDN-H and **matching Transformer performance under the n4k2 setting**, while maintaining strong extrapolation up to 16k tokens (with training context lengths ≤ 8k).
>
> | Model            | MMLU | MMLU-Pro | C-Eval  | AGIEval | BBH | DROP | GSM8k  | TriviaQA | Ruler (4k)  | Ruler (8k)  |Ruler (16k) |
> |------------------|-------|-------|--------|-------|-------|-------|-------|-------|-------|-------|-------|
> | Transformer      | 61.5 | 60.6 | 64.6  | 62.3 | **80.0** | 67.5 | 85.1 | 73.0 | **89.5**  | **81.5** | 0.0 |
> | GDN-H (1:6)      | 62.1 | 62.4 | 64.6  | 60.9 | 77.1 | 68.5 | 86.3 | 72.4 | 81.0  | 63.6 | 38.9 |
> | **SSE-GDN-H-n4k1 (1:6)** | 62.1 | 61.8 | 65.3  | 61.5 | 79.4 | **71.1** | 86.9 | 71.3 | 82.0  | 72.1 | 46.6 |
> | **SSE-GDN-H-n4k2 (1:6)** | **64.7** | **64.7** | **67.7**  | **64.3** | 78.3 | 69.1 | **87.6** | **73.2** | 88.5  | 81.2 | **48.7** |
>
> These results confirm that SSE extends effectively beyond diagonal linear attention and is also well-suited for delta-rule–based architectures.
>
> ---

---

> > ### Author Response · Authors · 2025-11-19
> > **[2/3] Response to Reviewer Jg4D (W2)**
> >
> > > **W2: Baselines seem to be cherry-picked since they are neither the most powerful nor fundamentally relevant sequence models.**
> >
> > Thank you for raising this concern. To further strengthen the credibility of our results, we have expanded our baseline coverage to include several widely adopted and recent architectures. (1) At the 600M-parameter, 15B-token scale, we trained Mamba, Mamba2, and KDA, and (2) at the 2B-parameter, 100B-token scale, we further evaluated Mamba2. The results show that **SSE continues to deliver the strongest performance**, particularly on recall-intensive tasks such as FDA.
> >
> >
> > 1. At the 600M-parameter & 15B-token scale, we trained Mamba [1], Mamba2 [2] , and KDA [3] (a recent linear attention method from Moonshot).
> >
> > | Model            | Wiki↓ | PIQA  | Hella. | Wino. | ARC-e | ARC-c | SIQA  | **CSR-Avg.** | -  | FDA   | SWDE  | SQuAD | **Recall-Avg.**  |
> > |------------------|-------|-------|--------|-------|-------|-------|-------|--------------|----|--------|--------|--------|------------------|
> > | Transformer      | 33.47 | 61.92 | 31.60  | 51.38 | 48.02 | 23.12 | 37.31 | 42.22        | -  | 74.50 | 60.67 | 32.67 | 55.95 |
> > | GLA              | 43.97 | 61.21 | 29.61  | 51.46 | 47.01 | 23.29 | 36.59 | 41.53        | -  | 9.44  | 23.40 | 23.06 | 18.63 |
> > | Mamba            | 36.62 | 63.82 | 32.20  | 50.43 | 49.62 | 23.38 | 38.59 | 43.01        | -  | 6.35  | 21.60 | 24.93 | 17.63 |
> > | Mamba2           | 34.89 | 62.89 | 32.84  | 48.93 | 49.20 | 23.81 | 37.92 | 42.60        | -  | 20.51 | 30.42 | 27.21 | 26.05 |
> > | KDA              | 34.76 | 62.19 | 31.81  | 51.22 | 49.96 | 24.83 | 37.41 | 42.90        | -  | 24.50 | 35.55 | 26.71 | 28.92 |
> > | GDN              | 35.26 | 62.79 | 32.16  | 51.85 | 50.46 | 24.23 | 36.80 | **43.05**    | -  | 14.79 | 33.48 | 26.24 | 24.84 |
> > | **SSE-n4k1 (Ours)**     | 35.52 | 61.59 | 31.71  | 51.62 | 49.45 | 24.74 | 38.33 | 42.91        | -  | 33.67 | 33.30 | 26.51 | **31.16** |
> > | **SSE-n4k2 (Ours)**     | 35.33 | 62.95 | 31.64  | 50.43 | 49.66 | 25.34 | 38.28 | **43.05**    | -  | 30.85 | 36.90 | 27.82 | **31.86** |
> > | **SSE-GDN-n4k1 (Ours)** | **34.15** | 63.11 | 32.22 | 50.36 | 50.59 | 23.98 | 37.46 | 42.95 | - | 41.56 | 41.13 | 30.83 | **37.84** |
> >
> > 2. At the 2B-parameter & 100B-token scale, we further conducted experiments for Mamba2.
> >
> > | Model        | Wiki↓ | PIQA | Hella. | Wino. | ARC-e | ARC-c | SIQA | CSR-Avg. | - | FDA | SWDE | SQuAD | Recall-Avg. |
> > |--------------|-------|------|--------|-------|--------|--------|-------|-----------|---|------|-------|--------|--------------|
> > | Transformer  | 16.46 | 71.82 | 52.70 | 58.80 | 63.76 | 31.31 | 42.89 | 53.55 | - | 86.48 | 82.99 | 49.53 | 73.00 |
> > | GLA          | 21.30 | 68.44 | 42.82 | 54.38 | 59.43 | 28.24 | 41.45 | 49.13 | - | 51.27 | 59.86 | 36.73 | 49.29 |
> > | Mamba2       | 17.27 | 71.87 | 53.34 | 57.06 | 64.52 | 32.59 | 42.17 | 53.59 | - | 56.62 | 63.28 | 43.06 | 54.32 |
> > | GDN          | 17.08 | 72.25 | 53.20 | 57.70 | 65.61 | 32.68 | 42.89 | 54.05 | - | 54.63 | 66.88 | 39.88 | 53.79 |
> > | **SSE-n4k1 (Ours)** | **16.92** | 71.98 | 53.62 | 60.14 | 66.67 | 32.94 | 42.07 | **54.57** | - | 71.23 | 69.94 | 43.20 | **61.46** |
> >
> > ---
> >
> > **Note:** For fairness, all models differ only in the attention mixer and do *not* include convolutional layers, except for the Mamba family, which inherently uses 1D convolutions and unified blocks.
> >
> > [1] Mamba: Linear-Time Sequence Modeling with Selective State Spaces. https://arxiv.org/abs/2312.00752
> > [2] Transformers are SSMs: Generalized Models and Efficient Algorithms Through Structured State Space Duality. https://arxiv.org/abs/2405.21060
> > [3] Kimi Linear: An Expressive, Efficient Attention Architecture. https://arxiv.org/abs/2510.26692
> >
> > ---

---

> > > ### Author Response · Authors · 2025-11-19
> > > **[3/3] Response to Reviewer Jg4D (W3)**
> > >
> > > > **W3: The paper uses hard partition selection (top-$k$) and softmax row selection, but the gradient treatment for the discrete top-$k$ is not described. How do gradients flow through Eqs. 7–9, and are any implementation tricks needed for stability?**
> > >
> > > Thank you for the question. Below we clarify how gradients propagate; the behavior matches the reference implementation in the supplementary file `mask_optimize_kernel.py`.
> > >
> > > 1. Gradients under hard partition selection (top-$k$).
> > > During the forward pass, a binary top-$k$ mask (or varlen index) ensures that only the selected $k$ partitions are updated in Eq. 9. In the backward pass of linear attention, we iterate over all $n$ partitions and compute gradients for $\mathbf{q}_t,\mathbf{k}_t,\mathbf{v}_t,\mathbf{e}_t$ only for the selected ones (Eqs. 9–10).
> > >
> > > - Gradients to $\mathbf{W}_q, \mathbf{W}_k, \mathbf{W}_v$.
> > >   Because the top-$k$ mask zeroes out unselected states, the gradients for $\mathbf{q}_t,\mathbf{k}_t,\mathbf{v}_t$ are non-zero only for the chosen partitions. However, unlike MoE, SSE uses shared attention parameters, so $\mathbf{W}_q, \mathbf{W}_k, \mathbf{W}_v$ receive the **sum** of gradients from all selected partitions. This avoids instability issues such as overly sparse parameter updates.
> > >
> > > - Gradients to $\mathbf{W}_e$.
> > >   The gradient $\partial e$ is sparse due to the hard top-$k$. However, after passing through the backward of softmax in Eq. 7, the resulting gradient for $\mathbf{W}_e$ becomes dense. With an appropriate balancing constraint (e.g., an auxiliary load-balancing loss), this gradient remains stable.
> > >
> > > 2. Gradients under softmax row selection.
> > > For the softmax-based feature map (Eq. 8), gradients first flow through the backward of Eqs. 9–10 into $\mathbf{k}_t$, and then through a standard softmax derivative (Eq. 8). Since no top-$k$ is involved here, gradients to $\mathbf{W}_k$ are dense and stable.
> > >
> > > 3. Implementation considerations.
> > > No additional tricks are required beyond the above. We only compute the forward and backward passes of the softmax in fp32 to maintain numerical precision. Thanks to shared parameters and the balancing loss, gradient flow remains stable even with hard top-$k$ selection.
> > >
> > > ---

---

### Official Review · Reviewer_dHEC · 2025-11-01

**Soundness:** 4
**Presentation:** 2
**Contribution:** 3
**Rating:** 6
**Confidence:** 4

**Summary:**

The paper proposes row-selector to bottleneck the update of state and use multiple partitions to expand state size for improved expressiveness. The results show their methods can get comparable performance with Transformers.

**Strengths:**

1. results are strong: SOTA in 2B reasoning model.
2. preliminaries are well-organized, proofs are completed.
3. enable multiple efficient parallelized implementations.

**Weaknesses:**

1. phrasing can be simplified and changed for better delivery: information classification, row-sparse -> row-selector. (TBH information classification is very confusing).
2. miss an important baseline Mamba/Mamba2

**Questions:**

1. can you provide some efficiency analysis, especially empirical evidence, when compared to baseline linear attention models?

---

> ### Author Response · Authors · 2025-11-19
> **[1/2] Response to Reviewer dHEC (W1~W2)**
>
> > **W1: The term "information classification" is confusing and does not clearly reflect the intended mechanism.**
>
> Thank you for the insightful suggestion. As you pointed out, the mechanism can indeed be regarded as a row-selection process. We will therefore reduce the emphasis on the term "classification" and refine the terminology to improve clarity for readers.
>
> In Section 3, we introduce the *row-sparse state update* framework, where the current input is assigned to a subset of state rows and only the selected rows are updated. Our original motivation for using the term *information classification* was to convey that different rows can be interpreted as latent categories, emphasizing the role of $f(\mathbf{x}_t, \mathbf{W}_k)$ as a classification function. This naturally motivates the *Softmax-top-$k$ hard classification* mechanism introduced in Section 3.2.
>
> In the revised version, we will preserve the conceptual intuition where needed, but describe the operation primarily as row selection to ensure clarity.
>
> ---
>
> > **W2: The paper misses important baselines such as Mamba/Mamba2.**
>
> Based on your suggestions, we have included several additional baselines to provide a more comprehensive comparison. The results show that **SSE continues to deliver the strongest performance**, particularly on recall-intensive tasks such as FDA.
>
> 1. At the 600M-parameter & 15B-token scale, we trained Mamba [1], Mamba2 [2] , and KDA [3] (a recent linear attention method from Moonshot). We also evaluated the combination of SSE with GDN, referred to as SSE-GDN (see our response to Reviewer Jg4D Weakness 1 for additional details).
>
> | Model            | Wiki↓ | PIQA  | Hella. | Wino. | ARC-e | ARC-c | SIQA  | **CSR-Avg.** | -  | FDA   | SWDE  | SQuAD | **Recall-Avg.**  |
> |------------------|-------|-------|--------|-------|-------|-------|-------|-------|-------|-------|-------|-------|-------|
> | Transformer      | 33.47 | 61.92 | 31.60  | 51.38 | 48.02 | 23.12 | 37.31 | 42.22 | -  | 74.50 | 60.67 | 32.67 | 55.95 |
> | GLA              | 43.97 | 61.21 | 29.61  | 51.46 | 47.01 | 23.29 | 36.59 | 41.53 | -  | 9.44  | 23.40 | 23.06 | 18.63 |
> | Mamba            | 36.62 | 63.82 | 32.20  | 50.43 | 49.62 | 23.38 | 38.59 | 43.01 | -  | 6.35  | 21.60 | 24.93 | 17.63 |
> | Mamba2           | 34.89 | 62.89 | 32.84  | 48.93 | 49.20 | 23.81 | 37.92 | 42.60 | -  | 20.51 | 30.42 | 27.21 | 26.05 |
> | KDA              | 34.76 | 62.19 | 31.81  | 51.22 | 49.96 | 24.83 | 37.41 | 42.90 | -  | 24.50 | 35.55 | 26.71 | 28.92 |
> | GDN              | 35.26 | 62.79 | 32.16  | 51.85 | 50.46 | 24.23 | 36.80 | **43.05** | -  | 14.79 | 33.48 | 26.24 | 24.84 |
> | **SSE-n4k1** (Ours)     | 35.52 | 61.59 | 31.71  | 51.62 | 49.45 | 24.74 | 38.33 | 42.91 | -  | 33.67 | 33.30 | 26.51 | **31.16** |
> | **SSE-n4k2** (Ours)     | 35.33 | 62.95 | 31.64  | 50.43 | 49.66 | 25.34 | 38.28 | **43.05** | -  | 30.85 | 36.90 | 27.82 | **31.86** |
> | **SSE-GDN-n4k1** (Ours)   | **34.15** | 63.11| 32.22| 50.36| 50.59| 23.98| 37.46| 42.95 | -  | 41.56 | 41.13 | 30.83 | **37.84** |
>
>
> 2. At the 2B-parameter & 100B-token scale, we further conducted experiments for Mamba2.
>
> | Model            | Wiki↓ | PIQA  | Hella. | Wino. | ARC-e | ARC-c | SIQA  | CSR-Avg. | -  | FDA   | SWDE  | SQuAD | Recall-Avg.  |
> |------------------|-------|-------|--------|-------|-------|-------|-------|-------|-------|-------|-------|-------|-------|
> | Transformer  | 16.46 | 71.82 | 52.70 | 58.80 | 63.76 | 31.31 | 42.89 | 53.55 | -  | 86.48 | 82.99 | 49.53 | 73.00 |
> | GLA          | 21.30 | 68.44 | 42.82 | 54.38 | 59.43 | 28.24 | 41.45 | 49.13 | -  | 51.27 | 59.86 | 36.73 | 49.29 |
> | Mamba2       | 17.27 | 71.87 | 53.34 | 57.06 | 64.52 | 32.59 | 42.17 | 53.59 | -  | 56.62 | 63.28 | 43.06 | 54.32 |
> | GDN          | 17.08 | 72.25 | 53.20 | 57.70 | 65.61 | 32.68 | 42.89 | 54.05 | -  | 54.63 | 66.88 | 39.88 | 53.79 |
> | **SSE-n4k1** (Ours) | **16.92**  | 71.98 | 53.62 | 60.14 | 66.67 | 32.94 | 42.07 | **54.57** | -  | 71.23 | 69.94 | 43.20  | **61.46** |
>
>
> *NOTE:* It is important to note that, to ensure a fair comparison, all of our models vary only in the attention mixer and **do not** include convolutional layers—except for the Mamba family, which inherently uses 1D convolutions and unified blocks.
>
> [1] Mamba: Linear-Time Sequence Modeling with Selective State Spaces. https://arxiv.org/abs/2312.00752
> [2] Transformers are SSMs: Generalized Models and Efficient Algorithms Through Structured State Space Duality. https://arxiv.org/abs/2405.21060
> [3] Kimi Linear: An Expressive, Efficient Attention Architecture. https://arxiv.org/abs/2510.26692
>
> ---

---

> > ### Author Response · Authors · 2025-11-19
> > **[2/2] Response to Reviewer dHEC (Q1)**
> >
> > > **Q1: Efficiency analysis especially empirical evidence compared to baseline linear attention models.**
> >
> > Thank you for the valid suggestion. We have expanded our efficiency study through two sets of experiments:
> >
> > 1. **Attention runtime (forward + backward, ms)**.
> > We benchmarked SSE (with GLA), GLA, and full attention from 8k to 500k sequence lengths.
> > **SSE exhibits linear scaling and becomes significantly faster than full attention beyond 32k**, while being moderately slower than GLA due to overheads such as sorting router indices, reordering Q/K/V/O, increased varlen segments (leading to more kernel-level chunk boundaries), and computing attention over $k{+}1$ state partitions per step.
> >
> > | Model / Seq Len            |  8k  | 16k | 32k | 64k | 128k | 256k  | 500k |
> > |------------------|-------|--------|-------|-------|-------|-------|--------------|
> > | Full Attention (Quadratic)   | 6 | 9  | 24 | 83 | 315 | 1250 | 4376 |
> > | GLA (Linear)                 | 3 | 5  | 10 | 18 | 36 | 71 | 131|
> > | SSE-n4k1-varlen (Linear)     | 8 | 14  | 26 | 50 | 97 | 191 | 358 |
> >
> > 2. **Training throughput (K tokens/s).**
> > We measured throughput on a single A100, using a 1.3B model for 2k–16k sequences and a 340M model for 32k.
> > We also evaluated SSE–GDN, the combination of SSE with the GDN architecture (see Reviewer Jg4D Weakness 1 for more details). The aggregated results are provided below.
> >
> > | Model / Seq Len & Batch Size            | 2k & 8 | 4k & 4  | 8k & 2 | 16k & 1 | 32k & 1 (340M params)|
> > |------------------|-------|-------|--------|-------|-------|
> > | Transformer++                | 24.24  | 22.35  | 19.41  | 15.25   | 25.14 |
> > | GLA                          | 19.55  | 19.47  | 19.44  | 19.23   | 53.01 |
> > | GDN                          | 21.37  | 21.07  | 20.87  | 20.48   | 55.71 |
> > | Mamba                        | 12.81  | 12.79  | 12.48  | 14.72   | 35.87 |
> > | Mamba2                       | 18.32  | 18.37  | 18.13  | 18.29   | 45.81 |
> > | MoM-GDN-n4k1                 | 11.12  | 11.23  | 11.16  | 10.20   | 23.71 |
> > | **SSE-GLA-n4k1**             | 12.40  | 12.33  | 12.29  | 12.29   | 28.79 |
> > | **SSE-GDN-n4k1**             | 14.88  | 14.86  | 14.73  | 14.45   | 35.08 |
> >
> >
> > Overall, **SSE is faster than Transformer++ at 32k**, while remaining slower than GLA/GDN/Mamba2 due to the inherent overhead of state expansion—**a clear speed–performance tradeoff**.
> > SSE–GDN reaches ≈70% of GDN throughput (similar to Mamba), and SSE–GLA reaches ≈60% of GLA. Despite this gap, SSE maintains sufficient training efficiency for large-scale long-context pretraining, and we view targeted low-level optimization as a further direction for narrow the remaining overhead.
> >
> > ---

---

### Author Response · Authors · 2025-11-25
**General Response 2025/11/25 (AOE)**

Dear ACs and Reviewers,

We sincerely thank all reviewers for the time and constructive feedback provided. Your comments have been invaluable in improving the clarity and completeness of our work.

Linear attention has received considerable attention as one of foundations for efficient large models, and hybrid attention architectures have shown promising performance in industrial pre-training settings. Nevertheless, a consistent performance gap remains between linear and full attention on tasks requiring strong recall, which in turn limits their reliability in long-horizon reasoning and constrains the attainable hybrid ratios during deployment. Our work aims to close this gap by developing more principled formulations of state updating and state expansion:

- **Deeper understanding of the state mechanism**: We conceptualize state rows as latent categories and introduce a top-$k$-then-softmax mechanism for sparse row selection.

- **An Efficient state expansion scheme**: Within this sparse row selection framework, we propose SSE, which scales the state capacity of linear attention while keeping both computational and parameter overhead controlled.

- **Comprehensive experimental validation**: Through full training pipelines and diverse benchmarks, we observe two key results: (1) SSE substantially improves recall performance, with gains that scale with increased state capacity (Table 1, Figure 5); (2) our 1:5 SSE hybrid model achieves state-of-the-art performance on post-training reasoning tasks (Table 4).

We hope that these findings offer additional insights for the development of linear attention mechanisms and demonstrate that improved recall can further strengthen efficiency in long-context scenarios.

&nbsp;

---

&nbsp;

In response to reviewers’ suggestions, we conducted extensive revisions and additional experiments:

- **Expanded baselines [Reviewers dHEC, Jg4D] and integration with GDN [Reviewer Jg4D]**: We added comparisons with Mamba, Mamba2, and KDA, and further evaluated the integration of SSE with GDN (scaling up to a 15B MoE model trained on 450B tokens). These results provide stronger evidence of SSE’s performance advantages.

- **More comprehensive efficiency evaluation [Reviewers dHEC, heMc, KhC1]**: We augmented runtime comparisons against GLA and full attention and added training throughput measurements. The results show that SSE achieves significantly higher efficiency than transformers on long sequences.

- **Writing and presentation improvements [Reviewers dHEC, heMc, KhC1]**: We revised the title [Reviewer heMc], terminology [Reviewers dHEC, heMc], logic flow [Reviewer heMc], figures [Reviewers heMc, KhC1], varlen implementation diagrams [Reviewer heMc], and related work [Reviewer KhC1] following the reviewers’ guidance. `An updated PDF has been uploaded, where main changes are marked in blue to facilitate efficient review`.

We have addressed each comment in detail in the individual responses. Should further clarification be needed, we would be happy to provide it.

---

---

### Author Response · Authors · 2025-12-03
**Summary to Area Chair**

**Dear Area Chair,**

We sincerely thank you and all reviewers for the time and effort dedicated to evaluating our paper. We truly appreciate the recognition of our work's strengths, including (1) the well-motivated methodology [Reviewers Jg4D, heMc, KhC1], (2) completed theoretical proofs [Reviewer dHEC], (3) solid experimental results [Reviewers dHEC, heMc, KhC1], and (4) efficient parallelized implementations [Reviewers dHEC, heMc, KhC1].

Throughout the rebuttal, we actively engaged with all reviewers and addressed each concern through (1) detailed clarifications, (2) expanded experiments spanning pre-training, downstream tasks, and efficiency, and (3) a revised manuscript with improved presentation. We sincerely thank **Reviewer KhC1 for raising the score to 8**, and we appreciate **Reviewer heMc's positive acknowledgment and decision to maintain the score of 8**.

For your convenience, we summarize our contributions and corresponding revisions below.

---

### **A Brief Summary of SSE**

Linear attention is a key building block for efficient large models, and hybrid architectures have shown promising performance for industrial pre-training. Nevertheless, a *consistent recall gap* limits their reliability in long-horizon reasoning and constrains the attainable hybrid ratios. Our work aims to close this gap with more principled state updating and state expansion:

- **Understanding state mechanisms**: We conceptualize state rows as latent categories and introduce a top-$k$-then-softmax mechanism for sparse row selection.

- **Efficient state expansion**: We propose SSE, which scales up linear attention state capacity while keeping both computational and parameter overhead controlled.

- **Comprehensive validation**: Through full training pipelines and diverse benchmarks, we observe two key results:
  - SSE substantially improves recall, with gains that scale with larger state sizes (Table 1, Figure 5).
  - Our 1:5 SSE hybrid model achieves *state-of-the-art* post-training reasoning performance (Table 4).

We hope these findings offer useful insights for advancing linear attention mechanisms and demonstrate that improved recall can further enhance efficiency, especially for the popular hybrid paradigm.

---

### **A Detailed Summary of Responses and Revisions**

In response to reviewers' suggestions, we conducted extensive revisions and experiments:

- **Expanded baselines** `[Reviewers dHEC, Jg4D]`: We added comparisons with Mamba, Mamba2, and KDA at the 600M-params/15B-token scale, and additional Mamba2 results at the 2B-params/100B-token scale. SSE consistently delivers the strongest performance, particularly on recall-intensive tasks.

- **Integration with GDN** `[Reviewer Jg4D]`: We conducted a thorough investigation of SSE-GDN and showed that SSE extends effectively to delta-rule-based architectures.
  - At the 600M-params/15B-token scale, SSE-GDN sets a new state-of-the-art on recall benchmarks and further validates the write–read gating design.
  - We further scale hybrid SSE-GDN-H to a 15B MoE model trained on 450B tokens, achieving substantial gains on general benchmarks and Ruler, matching Transformer performance.

- **Efficiency evaluation** `[Reviewers dHEC, heMc, KhC1]`: We augmented attention runtime comparisons against GLA and full attention (8k–500k lengths), and added training-throughput measurements for eight models (2k–32k lengths). SSE is significantly faster than Transformers on long sequences (>32k), provides sufficiently high throughput for large-scale training, and offers a clear speed–performance trade-off versus linear transformers without state expansion.

- **Gradient flow explanation** `[Reviewer Jg4D]`: We clarify gradient propagation under hard partition selection and softmax row selection. Parameter sharing and balancing loss ensure stable gradients.

- **Scaling experiments** `[Reviewer KhC1]`: We provide scaling results for 2B-parameter models trained on 20B/40B/100B tokens with a constant learning rate. SSE scales effectively: as training tokens increase, it gradually achieves the best performance across perplexity, commonsense reasoning, and recall metrics.

- **Writing and presentation improvements** `[Reviewers dHEC, heMc, KhC1]`: We revised the title [Reviewer heMc]; terminology (e.g., information classification) [Reviewers dHEC, heMc]; the logical flow (especially Section3) [Reviewer heMc]; figures and captions [Reviewers heMc, KhC1]; varlen implementation diagrams (Figure 8) [Reviewer heMc]; and related work [Reviewer KhC1]. An updated PDF has been uploaded, with main changes marked in *blue*.

---

Once again, we sincerely thank the AC and all reviewers. Your feedback is the strongest affirmation of our work's contribution to the community and has substantially improved the clarity and completeness of our paper. We hope that our responses fully address all concerns and support a positive reassessment of our work.

Best regards,

Authors of Submission8934

---

---

### Meta-Review · Area_Chair_d7AF · 2026-01-07

**Summary:**

This paper improves context compression in linear attention models with row-sparse update and sparse state expansion (SSE). Experiments on language modeling, needle-in-a-haystack and reasoning benchmarks show that SSE and the hybrid SSE-H achieves state-of-the-art performance among linear attention models. The reviewers proposed the following concerns:
1. The original manuscript missed the comparison on more recent linear-like attention work, such as Mamba/Mamba2, DeltaNet
2. The original manuscript lacks efficiency analysis such as wall time and scaling experiments.
3. The writing of the original manuscript may be largely improved.

**Reviewer Concerns:**

1. In the revised version, the authors added more experiments results, with comparisons on Mamba, Mamba3, GDN, showing SSE consistently improves the performance.
2. The authors also added results on attention runtime, training throughput on models from 340M to 1.3B, and scaling tokens from 20B to 100B, the concerns on efficiency and scaling experiments are mostly addressed.
3. The revision considers many of the review suggestions, making the new manuscript much better in presentation.

**Reviewer Scores:**

The original scores are 6(dHEC), 4(Jg4D), 8(heMc), 6(KhC1). Since most of the proposed concerns have been addressed, I would believe  an all-positive evaluation will be achieved, e.g. 6,6,8,8.

---

### Decision · Program_Chairs · 2026-01-26

Accept (Poster)